



# Rapid abiotic transformation of marine dissolved organic material to particulate organic material in surface and deep waters.

Paola Valdes Villaverde[1], Cesar Almeda Jauregui[1], Helmut Maske[1]

[1]CICESE, Carretera Tijuana-Ensenada No. 3918, Ensenada, Baja California, Mexico, CP 22860

*Correspondence to*: Helmut Maske (hmaske@cicese.mx)

**Abstract.**

Marine particulate organic matter (POM) supports the vertical transport of organics in the oceans, and the ecology of microbes and filter feeders. POM is collected on GFF filters and quantified as particulate organic carbon (POC) and nitrogen (PON). The filtrate contains dissolved organic matter (DOM) that partly is abiotically converted into POM and can be collected on

GFF filters. We filtered seawater from cultures and the pelagic ocean, after the initial sample filtration yielding the conventional POC/PON sample ($POM_1$), we refiltered the filtrate yielding $POM_2$, refiltering its filtrate we obtained $POM_3$ and so on till $POM_i$. Refiltered POM tended to the same concentration independent of the sample depth and even after a few refiltrations, independent of the original particulate organic load. $POM_2/POM_i$ for surface water (<100m) was about 0.1 and for deep water (>1000m) was >1.0. We considered adsorption of DOM and bacteria that had passed through the first filter but

concluded that these could not explain the high POM concentration in the filtrates. We suggest that POM in filtrates represent gels formed due to hydraulic stress during filtration. The ratio of $POM_2/POM_i$ will partly depend on environmental conditions like turbulent energy. We suggest that the increase of $POM_2/POM_i$ with depth is related to the lower in situ turbulent energy at greater depth. We discuss aspects of POM methodology, including problems with acidification of samples, and the wider ecological implications of our results.

## 1 Introduction

Particulate organic matter (POM) plays a central role in the biogeochemistry of the ocean, because the sinking particles fuel the vertical transport of chemical constituents, including the sequestration of atmospheric carbon, a process referred to as the biological pump or soft tissue pump. POM also plays a crucial role in the ecology of microbes and filter feeders from the surface to the deep ocean (Alldredge and Youngbluth, 1985). Organic particulate matter (POM), measured as particulate

organic carbon (POC) and particulate organic nitrogen (PON) in the ocean is conventionally defined by a lower size limit defined by the retention properties of glass fiber filters (GFF) that have a nominal cut-off size of 0.75μm, but after pre-combustion have a smaller particle size cut-off (Abdel-Moati, 1990; Nayar and Chou, 2003). We use the GFF filters to define the lower size limit of POM as common in oceanography, and POM is expected to be largely composed of detritus, fecal pellets, phytoplankton, prokaryote cells and cell aggregations. Most of these organic particles are contained by biological

membranes and are defined here as membrane enclosed particles (MEP). Their effective cut-off size and the particle density assures that most of the MEP retained by the filter is large enough to sink.





The method for POC and PON determination (CHN method) has been used largely unchanged for 50 years; the POM samples are filtered on pre-combusted glass fiber filters (GFF) without organic binder. The CHN instruments oxidize the sample

organics retained on the filter, separating the different gases on a chromatographic column, for example a thermal desorption column and then quantifying the gases ($CO_2$ and $N_2$) by thermal conductivity. In the literature different features of the POM methods have been discussed, from the best way to eliminate the inorganic carbon from the sample by acidification, how to obtain a representative POC sample from the water, e.g. Niskin bottles vs. in situ pump (Liu et al., 2009), how to avoid swimmers, or to account for the adsorption of dissolved organic matter (DOM) to the glass fibers of the filter. The adsorbed

organics on the filter (Maske and Garcia, 1994) have been treated as part of the sample blank of POC and PON using different approaches (Moran et al., 1999; Liu et al., 2005, Turnewitsch et al., 2007; Rasse et al., 2017; Novak et al., 2018). Despite the long history of the CHN method, these different aspects of the methodology are still largely unresolved.

Apart from MEPs there are hydrogels in the ocean that are bigger than the GFF filter pores and thus form part of the measured

POC and PON. Hydrogels or chemical fractions of it have many different names in the literature such as exopolymeric substances (EPS), self-assembling gels (SAG), mucilage, the perfect slime (Flemming et al., 2017), transparent exopolymer particles (TEPs) and Coomassie stainable particles (CSP; Long and Azam, 1996). In addition, marine snow is probably made up largely from gel-like substance (Alldredge and Silver, 1988), but we are not considering marine snow here because technically it would not be included in our data of refiltered samples. Already Riley (1963) noted the relative importance in

the ocean of organic aggregates, non-MEPs, and suggested that they are probably reversibly formed from dissolved organics. Parsons (1975) argued that there might be a dynamic balance between POC and dissolved organic carbon (DOC) in the sea based on similar temporal patterns during the year and similar $^{14}C/^{12}C$ ratios in the deep sea. Engel et al. (2004) suggested that polysaccharide aggregates played a significant part in the vertical carbon cycle of the ocean. We assume that hydrogels are partially excreted by phytoplankton (Ding et al., 2009) and bacteria, as reported for transparent exopolymer particles (TEP),

or they are formed from dissolved organic precursors. These are physical gels that form aggregates non-covalently through electrostatic or polar forces, initially in the form of colloids or nanogels that then continue aggregating to form microgels (Grant et al., 1973; Braccini and Pérez, 2001; Verdugo, 2007; Orellana and Leck, 2015). The microgels can then group to form larger aggregates that may include membrane-enclosed particles like cells and other detritus. These processes can be considered to form part of the general process of particle aggregation in the ocean as has been discussed by Burd and Jackson (2009). The

role of gels in the ocean have been investigated mainly by using the TEP methodology (Wurl and Cunliffe, 2017). The standard TEP method calls for membrane filtration with 0.45µm filter pores, similar to the effective cut-off size of pre-combusted GFF filters. Because of the similar pore size used in POC and TEP filtration, we know that the POC samples will include TEPs. TEPs represent an unknown proportion of all hydrogels and their carbon content is difficult to estimate (Villacorte et al., 2015). Of the many TEP related publications some were directed at their role in carbon cycling (Engel et al., 2004; Mari et al., 2017),

at the process of TEP formation and cycling (Wurl et al., 2011), and the interaction of TEPs with membrane enclosed particles,



MEPs (Kiørboe, 1997; Jackson, 1995). More recently, a different line of inquiries focused on marine hydrogels in general, and lately we learned that a significant part of organic carbon in the ocean is in form of hydrogels (Chin et al., 1998; Verdugo and Santschi, 2010; Verdugo, 2012; Orellana and Leck, 2015). Cisternas-Novoa et al. (2015) showed that TEP and CSP could be disaggregated by addition of EDTA, which probably chelates the bivalent cations, thus reducing the bivalent electrostatic

forces necessary to maintain the gel aggregates. This supported the general idea of abiotic formation of gel aggregates. Already Sheldon et al. (1967) suggested the spontaneous formation of organic particles in the deep sea. It is estimated that about 10% of the dissolved organic carbon (DOC) in the ocean is in the form of micro hydrogels in near surface samples (Verdugo, 2012). Considering that in the deep ocean the ratio of DOC to POC is >10:1 then this would suggest that the carbon contained in hydrogels in seawater is of the same order of magnitude as the bulk POC retained by filters. If we consider that the GFF filters

retain part of the hydrogels, then this suggests that hydrogels can form a significant part of the organics retained by the POC filter.

POM also includes part of the prokaryotes biomass. Because part of the marine prokaryotes community is smaller than the cut-off size of precombusted GFF filters, the relative proportion of prokaryote biomass included in POM will depend on the size distribution of the prokaryotes, and therefore on their physiological status. The prokaryote cells that passed the GFF filter

will partially be retained when a second GFF filter is placed below the top filter. The POC of the lower filter is normally interpreted as adsorbed dissolved organics but the organics measured on this second filter will include retained prokaryote cells that had passed the first filter. We propose that the oceanic POC measured by current protocols is made up partly by hydrogels and the dissolved organics that passed the GFF filters include precursors that then rapidly aggregate to form new hydrogels. The relative contribution of the hydrogels to the measured POC depends on the water depth. Because hydrogels

change much less with water depth than MEPs the contribution of gels to POM should increase with depth.

Hydrogels have different physical and chemical properties compared to MEPs, for example they are less rigid because they are not contained by membranes, gels are porous, containing large volume portions of water and thus have a specific weight similar to water making them less prone to sink and sometimes they can float (Mari et al., 2017). Furthermore, hydrogels have a higher carbon to nitrogen ratio because they consist mostly of carbohydrates. These differences should assign hydrogels a

specific ecological niche. The pelagic ecology can be strongly impacted by hydrogels because they form the glue that bond MEPs together, thus forming bigger MEPs-hydrogel aggregates that will show accelerated sinking rates (Burd and Jackson, 2009). The formation of gel aggregate has been related to the termination and sinking of phytoplankton blooms. It has been suggested that the hydrogel aggregation process is accelerated with higher turbulent energy. There is little information available on marine hydrogels except for a specific fraction, the TEPs. There is an abundance of information on filtered TEPs

but a shortage of information on total gel concentration in suspension, since they are difficult to detect optically by attenuation or reflectance without prior staining, and most stains are specific to the polymers composing the gel particle. The quantitative interpretation of filtered TEPs, or gels is challenging because of the spontaneous phase change as a result of the abiotic aggregation of precursors to micro-gels. We included here measurements of TEPs for comparison because, as the literature cited above shows the TEPs are by far the most researched fraction of marine gels. TEPs were shown to depend on the dynamic





balance between micro-gels and precursors which is defined by environmental conditions, and the chemical nature and concentration of the precursors. Below our data show the formation of significant amounts of micro-gels in filtrates on timescales that are too short for biological production thus suggesting abiotic precursor aggregation.

## 2. Methods

### 2.1 Samples sources and experimental strategy

Ocean samples were collected in the California Current, near the surface or at a depth of 950m in the California Current system, Station Antares (31° 45.000 N, -116°57.600 W). The deep samples were filtered through GFF after sampling and then aged in glass carboys in the dark for several months. The samples from the Gulf of Mexico were taken on two cruises of opportunity, one sample per day. Station locations, depths and concentrations are supplied in supplemental material. Because these were cruises of opportunity we were logistically limited to three sequential filtrations. Some experiments were performed with

cultures of *Thalassiosira weissflogii* (CCMP1336, NCMA Bigelow) and Pseudonitzschia sp. (local isolate by Dr. Ricardo Cruz Lopez). Cultures were maintained in natural seawater (aged deep water) with L1 media addition (NCMA Bigelow) at a temperature of 19° C and a 12h/12h light/dark cycle.

The experimental strategy (Fig.1) was to repeat filtration of samples through the same filter type (pre-combusted GFF filter (Whatman). POM samples were filtered on pre-combusted (450oC, 4 hours) GFF filters, 25mm (Whatman). We used

sequential vacuum filtration (Fig. 1) with careful control of differential pressure (5 kPa). The first filtration (POM1) would be equivalent to a normal POC and PON sample (POC1 and PON1). The sample-type number corresponds to the number of sequential filtrations of the same sample. The sequential vacuum filtration implied that one filtration step had to be terminated for the filtrate to be available for the next filtration step, this determined the minimum time between each filtration step to be about 1 hour. This minimum time delay between successive filtrations (F1 to Fi) varied depending on the sample volume, if

double filters were used to check adsorption, and also on the number of parallel filtrations of different samples.

### 2.2 Particulate organics measurement

Sample filters were stored in pre-combusted glass scintillation vials at -20 oC and dried for 12 hours or more at 60 °C (Rosengard et al., 2018). POC and PON samples were measured with a MicroCube instrument (Elementar) calibrated with Acetanilide (CAS 103-84-4, Merck Darmstadt) at the beginning of the sample run and after approximately every 10th sample.

Standard concentrations were close to the concentration of the samples. We found no tendency of a change in standard slope during sample runs. Including all standards during a run, typical correlation coefficients ($R^2$) were 0.99. Regression slopes were similar even after repacking the oxidation and reduction columns. For instrument blanks we used dry precombusted GFF filter for blanks. Sometimes a second filter was introduced below the top sample filters to estimate the adsorption of DOM to the filter fibers.



The data of actual samples were less reproducible than standards (Fig. 2), possible reasons may have been sample heterogeneity, time periods between filtration steps, differences in individual filters or hydraulic shear exposure during filtration. The variance does not seem to increase with concentration (homoscedastic behaviour) for either POC or PON, meaning that the coefficient of variation decreases with concentration. We can use two times the standard error of the intercept to estimate the 95% range of values falling within +/- 0.51 μmol POC L$^{-1}$ and +/- 0.05 μmol PON L$^{-1}$. Actually, if we use the

12 non-replicate POC and PON values from greater than 1000m depth (Fig.5) and exclude one outlier, we arrive at standard deviations of 0.50 μmol POC L$^{-1}$ and 0.017 μmol PON L$^{-1}$. This means that the reproducibility of the method is similar to the concentration reproducibility o deep samples and we cannot distinguish between different samples.

After initial trials we decided not to treat our samples with acid because a) we found the traditional acid fume exposure method to be faulty (see below), b) we considered it unlikely that refiltered samples contained inorganic carbon particles, and c) the

literature suggests that gels are sensitive to the pH and tend to disintegrate at low pH. We considered the possibility of a low concentration of inorganic carbon in $POM_1$ samples but we did not want to use a method for $POM_1$ samples that was different from $POM_2$ to $POM_i$ samples.

We experimented with acid vapor exposure of blank precombusted GFF filters similar to common sample treatment. At the bottom of a glass desiccator, a glass vessel with 11N HCl was placed. The filters were wetted with 0.5 or 1 ml of distilled

water and placed in precombusted scintillation vials. Exposure to acid fumes was 24 hours within a glass desiccator. Blank filters were wetted and placed inside precombusted vials before being placed together with the acid treated filters in a drying oven (60° Celsius). Initially for two experiments, the glass desiccator was sealed with silicon high vacuum grease (Dow Corning). For our experiments with grease free acid vapor treatment, we thoroughly cleaned the glass desiccator initially with acetone, in another experiment with detergent and in a third experiment again with acetone. For all experiments dry

precombusted GFF filters were used as blanks.

For the processing of Micro Cube instrument data we reviewed all standard and sample peak integrations individually, we then subtracted from the sample peak a blank value obtained by wetting a precombusted GFF filter with 1 mL of sample and otherwise treated the same as the samples. To the difference between sample and blank we applied the regression slope of the acetonitrile standards to calculate the organic carbon and nitrogen of the samples.

**2.3 Transparent Exopolymer Particles (TEPs)**

We used the colorimetric method based on Passow and Alldredge (1995) to quantify TEPs (Hakspiel et al, 2017), Samples were collected on 0.45 mm polycarbonate filters (Poretics), stained with 0.02 % Alcian Blue in 0.06 % glacial acid, and dissolved in 80 % sulphuric acid. The spectral peak at 787nm was used to calculate the TEP concentration using a calibration curve of Xanthan Gum (XG) and a TEP carbon equivalence to Xantham Gum of 0.63 gC gXG$^{-1}$.





## 2.4 Bacterial counts

Bacteria were fixed with buffered formaldehyde (2 % final concentration) before storage in the refrigerator. The sample was incubated with 4′,6-diamino-2-phenylindole (DAPI, final concentration 1 µg mL−1) (Porter and Feig, 1980) and filtered immediately on 0.2 µm black polycarbonate filters (Poretics). A total of more than 300 cells were counted for each sample using a Carl Zeiss epifluorescence microscope with an X100 objective, and a 175W xenon lamp (Lambda LS, Sutter) connected through a liquid light guide.

All regressions presented here were of type 2 (Pearson) except where noted. For the comparison of data sets, we used the Mann-Whitney test considering non-normal distribution (Statistica 7.1). No outliers were excluded in the data analysis except when specifically noted.

## 3 Results

### 3.1. Acid pretreatment of samples to eliminate inorganic carbon

We investigated the effect of vapor acid exposure to samples to eliminate inorganic carbon. Humidified precombusted GFF filters were exposed to HCl fumes in a glass desiccator (Yamamouro and Kayanne, 1995). The ground glass joint of the desiccator top was sealed dry (experiments 1, 2 and 3) or with silicon high vacuum grease (Dow Corning), experiments 5 and 6. In experiment #3 (Fig. 3) two acid treated filters were accidentally joined in measurement, here we report only one value of half the POC and PON measured. In experiment #5 the HCl was more than 0.5 years old. Experiment #6 used new HCl.

POC in Figure 3, yielded significant differences (p<0.05) between A and B in experiments #2, 3, 5 and 6. POC #1 showed no significant difference between A and B samples (p>0.05). Average POC values: #1: A= -0.05 µmoles, B= 0.14 µmoles; #2: A= 1.31, B = -11 µmoles; #3: A= 0.23, B= -0.06 µmoles; #5: A= 2.44, B= -0.02 µmoles; #6: A= 1.39, B= -0.03 µmoles. Four of the no-acidified samples showed negative POC values. The average POC of all non-acidified samples was -0.017 µmoles, i.e. the distilled water wetted GFF samples had on average lower values than the dry GFF used to define the zero. The PON data in Figure 3 showed no significant differences (p>0.05) between acidified (A) and non-acid exposed (B) samples. The average PON of all non-acidified samples was 0.012 µmoles,

### 3.2. Effect of DOC adsorption to a 2nd filter

In part of the experiments we used the 2-filter method to estimate the DOM absorption as has been done before (Menzel, 1967; Turnewitsch et al., 2007; Liu et al., 2005, 2009; Cetinç et al., 2012). The underlying concept is that the filtrate that had passed the first filter contained only DOM and all organic carbon measured on the second filter underlying the top filter would be adsorbed DOC. The organic carbon of the second filter could then be subtracted from the top filter to correct the top filter sample for the organics adsorbed and thus yield the concentration of organic particles. This experimental setup had only been used for the first filtration step before. Here we repeatedly refiltered the filtrates through a double layer of filters.





The data in Figure 4 clearly show diminishing presence of organic particles in top filters even after repeated double-filter filtrations. The POC retained by the bottom filter was lower than the top filter even after several refiltrations. In Fig. 4C the POC/PON ratio of the top filter was initially below the ratio of the bottom filter but similar after the second filtration. The geometric means of the POC/PON ratio for top filter, was 14 for the 1st filtration and 21 for the 2nd to 5th filtration. The POC/PON ratio for the bottom filter was 130 for the 1st filtration, and 53 for the 2nd to 5th filtration. This change in ratio would be consistent with a reduction in biomass and increase in hydrocarbon gels.

We normalized the data in Figure 4 by dividing the POC of the bottom filter by the POC of the top filter. In Figure 4D the geometric means of the bottom/top filter ratio for $POC_1$ is 0.11 and for $POC_2$ it is 0.17. With further refiltrations the ratio is trending towards 1.0.The data suggest that organic particle in form of gels are formed in the filtrate and retained at the next refiltration. One possible explanation is that the aggregation process is helped by the shear stress when the sample is passing through the GFF filter. In this case the POM measured on the underlying second filter could also partly originate from the aggregated DOM that had passed the top filter.

### 3.3. Sequential filtration of samples

In Figure 5 we show the general pattern of POC and PON, and their ratio with refiltration through single filters. As expected the POC and PON concentrations for the initial filtration ($F_1$ ) are significantly lower for the mesopelagic and deep samples than the surface samples: <100m samples: POC, PON respectively 10.06, 1.22 μmol L$^{-1}$); >1000m samples: 0.50, 0.017 μmol L$^{-1}$.The refiltered samples of <100m show a decrease with each refiltration for POC and PON. Using the geometric means, the ratios were 0.31 = 0.96 $POC_2$/ 3.09 $POC_1$, and 0.11 = 0.035 $PON_2$ / 0.33 $PON_1$. Neither $POC_2$ nor $PON_2$ were significantly different for the depths <1000m (p>0.05). The $PON_3$ and $POC_3$ values are similar to $POC_2$ and $PON_2$, converging with further refiltrations on concentrations of 1μm L$^{-1}$ POC and <0.1 μm L$^{-1}$ PON. For samples >1000m depths the POM$_2$ was significantly greater than POM$_1$ (p<0.05). The geometric means of POC1/PON1 ratio for samples 0-100m: 8.27, for 100-1000m: 11.3, for >1000m: 21.9. The ratios of the mesopelagic and deep samples are significantly different (p<0.05) from the ratio of the surface samples indicating less live organic matter (Figure 5 E and F). The geometric mean of refiltered POC2/PON2 ratios for 0-100m: 24.3, 100-1000m: 23.8, >1000m 18.1 are not significantly different and similar to the POC1/PON1- of samples from >1000m. When the data are normalized (Figure 5F) to the ratio of the first filtered sample then the pattern found in Figs. 5B and 5C is not clearly repeated. To summarize the tendencies in Figure 5: In the near surface samples (0-100m) the POC and the PON decreased by a factor 10 with the first refiltration whereas in deep samples (>1000m) the first refiltration yielded a relatively higher particle concentration, but the refiltered POC and PON concentrations from the different depth ranges is similar

Figure 6 shows the same data as Figure 5 but as POM$_2$/POM$_1$ ratios for carbon (Figure 6a) and nitrogen (Figure 6b) versus depth. Both figures show significant depth dependence (p<0.05); Figure 6a: log($POC_2$/$POC_1$) = 0.488 log(depth) -1.132, R$^2$: 0.68; Figure 6b: log($PON_2$/$PON_1$) = 0.572 log(depth) -1.391, R$^2$: 0.51.



### 3.4. Transparent extracellular particles, TEPS

Transparent extracellular particles are hydrogels and therefore we expected to find a similar pattern with refiltration as for
225 POC and PON, specifically because both methods use filters with similar lower particle size cut-offs. The initial concentrations
($TEP_1$) in Figure 7A are in the expected range for filters of $0.45\,\mu m$. In contrast to Figure 5, the different depth ranges did not
show the same clear differences for the first filtration. All the samples show significant TEP concentrations when the filtrates
are refiltered. The 0-100m samples showed a tendency for $TEP_2/TEP_1$ to decrease (p < 0.005), but this was not found with
deeper samples. The continued presence of TEP in the refiltered samples demonstrates that TEPs can be formed in the filtrates.
230 In Figure 7B two samples show a strong increase from $TEP_2$ to $TEP_3$, these two samples were kept in the refrigerator during
the night to limit bacterial activity, but the low temperature probably promoted the gel aggregation. The formation of TEPs in
the filtrates should be an abiotic process because the refiltration of $TEP_2$ was within about one hour and the presence of bacteria
was reduced after the initial $TEP_1$ filtration. It is known that TEPs exist in a size continuum because when filters with smaller
pore size are used to collect TEPs than significantly more TEP is measured (e.g. Hakspiel et al., 2017). The existence of the
235 smaller size class TEPs can then serve as precursors to form the bigger TEP aggregates that are then collected by the
conventional pore size of 0.45 μm.

### 3.5. The bacterial abundance in refiltered samples

Bacterial biomass should contribute significantly to the POC retained when filtrates are filtered. Lee et al. (1995) reported that
35 to 43% and 49% of the original bacterial concentration passed GFF filters. Figure 8 shows that a small and with successive
240 filtrations diminishing percentage of bacteria are retained by the GFF filter. Our bacterial abundance retention with successive
filtration is given for double precombusted GFF filters by Eq. (1) and for single GFF filters by Eq. (2).

$(B_i/B_0) = 1.066\ (F_1 +1)^{-0.487}$ ;  $r^2$: 0.88, n: 18.      Eq. 1

$(B_i/B_0) = 0.8422\ (F_1 +1)^{-1.341}$ ;  $r^2$: 0.95, n: 5.      Eq. 2

245

It is notable that the double filters did not collect the bacteria per filter as efficiently as single filter filtrations. The first set of
double filters reduced the bacterial concentration only about 32 percent, whereas the first single filter reduced the concentration
74 percent. The bacterial retention efficiency reported by Lee et al. (1995) for the same filter type was 65 to 51%. Lee et al.
(1995) did not use pre-combusted filters which might mean that the effective pore size was bigger than for our pre-combusted
GFFs. Their retention efficiency lies between our double filter and single filter efficiencies. The bacteria retention efficiency
of the first filter, single or in the form of double filters, will depend strongly on the average size of the bacteria and hence on
their physiological status. For example, it might be expected that deep ocean bacteria are less efficiently retained by the first
filter. As Figure 8 shows, the subsequent filtration steps retain only a few percent of the original bacterial population. It is
interesting that the relative filtration efficiency is relatively reduced with each step probably as a result of size selection, with



each step the cells get smaller and are retained less efficiently. For the current manuscript it is important that after the initial filtration step only a small percentage of the initial population is retained on the filters and thus can contribute only little to samples from $POM_2$ to $POC_i$.

## 4. Interpretation

Our data suggest that the sample as it passes through the glass fibre filter is exposed to hydraulic stress such as fluid shear, that
is responsible for the aggregation of dissolved organics into gels (gel HySt). This aggregation process has to take place in a very short time; assuming that the liquid volume within the filter is 1mL and the flow rate through the filter is 1L/hour, then the sample liquid would only be exposed to the hydraulic stress for 4 seconds. Our interpretation depends on a careful consideration of the different processes that are involved in the collection and measurement of the organics attached to the filter. Despite the widespread measurement of pelagic marine POC and PON (Martiny et al., 2014) there are still unresolved
questions about the proper method and the interpretation of the data. Specifically, when POM concentrations are low in deep-water samples, the data interpretation hinges largely on methodological details such as acid treatment, the correction of the POC for dissolved organics adsorbed to the filter surface, or the contribution of bacteria to POM. When dissolved and particulate fractions are defined by GFF filters, then bacteria are present in both fractions. The residual bacteria in the filtrates can mislead the interpretation of refiltered POM, that is why they have to be considered here. In general, there is a certain
ambiguity in the distinction between particulate and dissolved organic fractions in the ocean, partly because organics show a continuous size spectrum and the cut-off size limit in our work was given by methodological limitations, but also because of the exchange of organics between both phases. Already Riley (1963) showed that DOM can be transformed into POM by bubbling filtered surface seawater, yielding approximately 1/3 additional POM relative to the POM in the unfiltered sample. Recently Robinson et al. (2019) demonstrated that bubbling of filtered seawater led to TEP formation from DOM. If gels
constitute a significant part of the POM and these gels can be formed rapidly and abiotically from the dissolved precursors, then POM is largely defined by the balance between precursors and gels and the conditions that control the kinetics of this balance. Near the ocean surface where membrane enclosed particles are more concentrated the gels play a minor role and the relative POM in the refiltered samples is not dominant. Ortega-Retuerta et al. (2019) measured POC and TEPs at different depth and different regions inside and outside the Mediterranean. Their ratio of TEPs to POC decreased 20 to 30% percent
with depth, but this change might be specific for TEPs and not for all hydrogels. We would expect the proportion of gel-like organics to POC to increase with depths considering that POC actually increases at the first refiltration ($POM_2$) (Fig. 6).

Our data show in general the presence of particulate organic material even after several refiltrations. There are two basic explanations; one, that the filters at each filtration step capture with a certain probability only part of the particles that could be retained by the filter, i.e. bacteria pass the filter with a certain probability at each filtration step; two, that new particles are
formed in the filtrate. Assuming that at each filtration only part of the bacteria was retained then the bacterial abundance would decrease with sequential refiltrations. But the retained bacterial biomass is not sufficient to explain all the POM encountered and therefore our data demonstrate the formation of new hydrogels in the filtrates. The formation of hydrogels in the filtrate





cannot be explained by a phase transition between the collapsed to a swollen state of the gels (Tanaka, 1992), because this transition would only follow a change in sample conditions, for example temperature, pH or ionic concentration, changes that did not happen in the filtrates. Below we interpret our data along the two lines, methodology and ecological implications.

## 4.1. Methodological interpretation

### 4.1.1. Acidification of POC samples

Generally oceanic POC samples are pretreated with acid to eliminate the inorganic particulate carbon. The two methods commonly used are either adding a very small volume of diluted acid to the sample filter, or exposing the sample to acid vapors. There is no consensus about the best method but the latter method is most commonly employed. Publications indicate that for sediment samples the popular acid-vapor method can introduce systematic overestimation (Brodie et al., 2011) and errors in isotopic composition (Schubert and Nielsen, 2000). The latter two publications trace the measurement errors to silicon greased glass desiccator tops used for HCl vapor treatment. Our results show that the acid-vapor treatment using a silicon grease sealed glass dessicator increased the measured POC even in filtered plankton samples, but did not change the PON (Fig. 3). Without using grease, we found some or no increase in POC (Fig. 3). A comparison with global POC data (Martiny et al. 2014) shows that the POC added by acid treatment could significantly increase the POC concentration, especially for samples below the epipelagic layer. The relative POC concentration increase would depend on the filtered sample volume, which makes it difficult to correct the POC concentrations reported in data banks without detailed methodological information. Together with the POC, the POC/PON ratio would increase because the PON showed no increase with acidification. We suggest that without greased dessicators tops the acid vapor method would improve but our results do not absolve this method from a possible increase POC concentration.

We took the results in Figure 3 as justification for not having acid-treated our samples. Other reasons included the unlikely possibility that refiltered samples contained inorganic carbon particles; that gels are possibly sensitive to the pH and tend to disintegrate at low pH, and that acidification of $POM_1$ samples would result in two different methods for $POM_2$ to $POM_i$ samples.

### 4.1.3. Adsorption of organics to the filter surface

One of the methodological artifacts in the POM methodology is the tendency of dissolved organic material to adhere to inorganic filter surfaces (Maske and Garcia, 1994; Moran et al., 1999; Gardner et al., 2003; Rasse et al., 2017) and thus adding a small fraction of dissolved organic matter (DOM) to the POM sample. This adsorbed DOM would form part of the POM blank that has to be subtracted to obtain true POM. Recently Novak et al., (2018) investigated POC retained on GFF filters when different volumes of pre-filtered sample were filtered. He adjusted a global non-linear model suggesting that it would help to correct POC data that were previously not corrected for adsorbed DOC. His results are consistent with our argument that the pre-filtration had caused a phase transition of DOC to POC. For POC previously reported blanks were 1.25 to 1.75



$\mu$mol C sample$^{-1}$ (see references in Rasse et al., 2017). Incidentally, considering the results in Figure 3, this would be about

the amount of POC concentration added to acid vapor treated oceanic samples if one-liter sample size is considered. Rasse et al. (2017) used two different methods to estimate the contribution of adsorbed DOM, a) filtering different volumes of the same sample and assuming that the amount of adsorbed DOC is independent of the volume filtered and thus indicated by the POC axis intercept; b) refiltering the sample. The latter approach has been discussed by Turnewitsch et al. (2007) and is recommended by the U.S.JGOFS protocol (http://usjgofs.whoi.edu/eqpac-docs/proto-18.html). The refiltration of the sample

filtrate for a blank is similar to our approach to estimate the gels formed in the filtrate. Rasse et al. (2017) found that, as the volume of filtrate that was refiltered increased, the measured POC on the filters increased. They did not consider the collection of bacteria or the formation of hydrogels to explain this pattern, but used the POC of an intermediate refiltered volume to indicate adsorbed DOC to the filter. According to this interpretation they found for one cruise that method (b) yielded higher values than method (a), and in another cruise the results were similar. Our results suggest that the POC encountered in refiltered

samples has only minor relation with adsorbed organics but is the result of gel aggregation.

We consistently found a small amount of organics in a second filters placed directly underneath the top, primary POM filter. It might be argued that these organics are bacteria that had passed through the first filter (see below) or that they are gel aggregates that are rapidly formed from dissolved organics in the interface between both filters, similar to our $POM_2$ and subsequent refiltrations. One problem with this interpretation is the very short time available during the filter passage for the

gel-precursors to form aggregates. Maske and Garcia (1994) immersed filters in $^{14}$C enriched dissolved organic matter and found that inorganic filters, glass fiber or aluminum oxide, always adsorbed more $^{14}$C than filters made of organic material. Because they did not pass, the sample through the filters there was no collection of particles on the filter. The collection of $^{14}$C-organics on the filters was within a few seconds and without turbulence that might provoke the formation of gels. Maske and Garcia (1994) interpreted these result as adsorption driven by polarity or electrostatic effects. Therefore, we assume that

in our present data (Fig. 5) where the geometric means of the POC ratio of bottom to top filters was 0.11 for $POC_1$ and 0.17 for $POC_2$ is partially the result of adsorption.

### 4.1.4. The contribution of bacteria to the POC of refiltered samples

Apart from the adsorption of organics or hydrogel formation, another possible explanation for POM in secondary filters is the retention of bacteria that had passed the top filter and then are trapped with a certain probability in a second filter. A simple

conceptual model would consider three processes in a depth filter like GFF, which is really a maze with a range of different channel sizes: 1. Particles are retained because they are too big for the dominant pore size. 2. Smaller particles pass through the bigger channels of the filter maze with a certain probability depending on the maze architecture. 3. These smaller particles get trapped in the smaller channels of the lower filter. This conceptual model is similar to the functioning of size exclusion chromatography except that in the case of GFF filters the maze is made of glass fibers and in SEC by gels. It might be possible

that gels trapped in the filters help to retain some of the small particles, for example bacteria.





In Fig. 9 we try to estimate the contribution of bacterial biomass to the POC measured on GFF filters. Every POC sample will include some bacterial biomass depending of the bacterial abundance, cell size and filter retention efficiency. The contribution of bacteria to the POC (Fig. 9) is calculated based on counted bacterial abundance of the sample, the filtrates and assuming 12.4 fg carbon cell$^{-1}$ (Fukuda et al., 1998). The estimated bacterial carbon has a wide margin of error because the retention

efficiency of bacteria on GFF filters depends on the cell size, and possibly on the gel quantity that might trap bacteria. Cell size depends on the trophic status of the bacterial population, but it is expected to shrink with successive filtrations including in the case of stacked filters that are used to estimate the adsorption of DOM to the filter surface. The bigger cells will be trapped in the first filters and for the smaller cell that pass the first filter the second filter will have a lower retention efficiency. In Figure 8 we used 50% retention of bacteria at every GFF passage. This assumption follows roughly the bacterial abundance

data in Figure 8. We have no information about the impact of a decrease in bacterial cell size with successive filtrations, but we can assume that the retention efficiency decrease with each filtration step because of the smaller average cell size, also because cell size decreases the contribution of bacterial POC cell$^{-1}$ will decrease. Therefore, we expect that the bacterial POC in Figure 8 is overestimated for refiltered samples.

A comparison of the 1:1 line with the data (Fig. 9) shows that the bacterial biomass contributes little to the measured $POC_1$,

most samples show much higher POC values than the estimated bacterial POC. For refiltered samples, that is bottom filters or $POC_2$, up to $POC_i$. only a few measured POC data approached the 1:1 line (Fig. 9). As pointed out above, the bacterial biomass of these samples is probably overestimated. We interpret the difference between the 1:1 line and the refiltered samples as gels that were formed abiotically in the filtrate. Physiological bacterial gel production during our experiments is not expected to be significant because a) The time between the filtration steps was on the order of one hour, too little time for biogenic production,

b) In the filtrate, after the initial rapid formation of POC the particulate organics do not increase, and c) further refiltration did not decrease much the formation of the particles although the concentration of bacteria was reduced each time.

We conclude that the bacterial biomass does contribute to measured POC but does not explain the high POC measured in refiltered samples. The difference between bacterial POC and measured POC we interpret as abiotically formed gels by aggregation from precursors. Gel formation is particularly intriguing in >1000m deep samples where the $POC_2$ is mostly higher

than $POC_1$ samples (Figure 5A, B). We suggest that this increase is the result of gel-aggregation induced by shear stress during the passage of the sample through the filter maze. The unexpected increase of $POM_2$ over $POM_1$ could not be related to a change in physical and chemical properties of the sample because these stayed constant. $POM_3$ of deep samples is generally lower that $POM_2$ (Figure 5), probably because the concentration of precursors is reduced.

### 4.1.5. Transparent extracellular particles, TEPS

The marine gel fraction that has received most research are TEPs. TEPs are mainly interpreted as exudates of photosynthesizing phytoplankton although they can be found throughout the water column. TEPs are a fraction of marine gels and are expected to show similar aggregation patterns to the hydrogels in general. Abiotic aggregation of TEPs from precursors can be increased by turbulent energy (Passow, 2000; Prieto et al., 2001; Burns et al., 2019). Even bubbles can increase TEP formation (Johnson



& Cooke, 1980; Robinson et al., 2019) a process that is related to the technique of foaming-out used in water treatment plants.
The dependence of TEPS on turbulent energy makes the interpretation of TEP concentration more difficult because theoretically the prefiltration turbulent history should influence the concentration.

Another complication with the interpretation of the TEP concentration is due to the compound Xanthan gum (XG) a bacterial exopolysaccharide that is used as a standard to quantify TEP. Xanthan gum is not a typical component of dissolved or particulate organics in the ocean. Engel and Passow (2001) experimentally arrived at a relationship between TEP carbon and
XG of 0.75 µg C (µg XG)$^{-1}$. A comparison of concentrations has to consider the pore size of the collecting filter, 0.45µm and the difference in specific staining between XG and natural marine gels, i.e. the color formed per organic carbon. The importance of the filter pores can be shown by comparing different filters; Hackspiel et al. (2017) measure a median concentration increase of a factor of 4.1 in TEPs when the standard filter of 0.45 µm was switched to 0.22µm. For a comparison of concentrations, we can assume that $POC_2$ consists of gels, and compare $POC_2$ with the carbon content of Gum Xanthan
equivalent of $TEP_1$. When we apply the conversion factor of proposed by Engel and Passow (2001), then this comparison yields a 3.7 times higher carbon content for $TEP_1$. Considering the concentration difference of 4.1 between $TEP_1$ filtered with 0.22 µm and 0.45 µm (Hakspiel et al., 2017), then the difference in carbon content between $POC_2$ and $TEP_1$ might be partly explained by the pore size difference between POC and TEP samples (nominally 0.75 µm versus 0.45 µm). Considering that TEPs are composed of a specific fraction of dissolved organics, we expect that TEPs represent only a fraction of the organic
carbon of gels and therefore should contain significantly less carbon than the total gel. This is not the case; the high carbon estimate for TEPs might be partly explained by the differences in staining properties between Gum Xanthan and natural gels. TEPs did behave similar to POC in that refiltration resulted in significant concentrations of TEPs because of reassembly of dissolved precursors in the filtrate (Figure 7).

### 4.2. Biogeochemical interpretation

Parsons (1975) argued that there might be a dynamic balance between POC and DOC in the sea based on two observations, similar temporal patterns during the year and similar 14C/12C ratios in the deep sea. These observations could be interpreted either way, that the organisms are responsible for producing DOC or that the DOC is producing POC; both views would support Parsons observation. In general, we can expect that physical gels that are based on electrostatic interaction, as conceptualized in the egg-box model (Grant et al., 1973) or linked by polar forces allow for a faster transition between the
dissolved and gel state than the chemical gels that are based on covalently ties between precursors. Burd and Jackson (2009) discussed the process and ecological consequences of the aggregation of gels and membrane enclosed particles in the ocean. Guo et al. (2000) showed experimentally that in a continuously pumped tangential filtration system the membranes would retain molecules smaller than the nominal size cut off. They concluded that the unexpected retention would lead to an overestimation of the colloidal fraction. It should be pointed out that Guo et al (2000) worked on a much smaller size scale
than the present work, but the general trend was similar. Our data support rapid and non-biogenic phase transition from



dissolved to gel state, based on the quantity of POC produced in sequential filtrates. In our deep water samples (>1000m) the gels collected after the first filtration ($POC_2$) are typically more than the membrane enclosed particles and gels collected with the first filter ($POC_1$), which is the particulate organic carbon reported in the literature. In surface waters, the ($POC_2$) is typically lower than the ($POC_1$). With further refiltrations $POC_3$ and higher the organic material collected by the filter tends

towards a concentration of 1 µmol C L$^{-1}$ independent of the depth of the sample. We propose that this organic material is made up of gels that reassembled after the filtration. If precursors represent a significant part of the DOC in deep waters which has approximately 40 µmoles C L$^{-1}$ and that only part of the DOC consists of precursors, then with each refiltration only part of the precursors will be removed. After the initial refiltration there should still be sufficient precursors available in the DOC fraction for the gel formation during several refiltrations.

We tried to eliminate other processes that might interfere with measured POC concentration, like the acid fume treatment of samples, the adsorption of DOC and the bacteria that are found in filtrates. We found that none of these processes could explain the POC in filtrates satisfactorily, specifically not the relative high increase in POC in samples from >1000m depth. Already Passow (2000) showed that one fraction of gels, the TEPs could be formed abiotically with turbulence from precursor solution derived from surface waters or phytoplankton cultures that had passed a 0.2 µm filter. She used a Couette cell to treat the

precursor solution for 24 hours with a defined level of turbulent energy. At the end of the experiment, she always found a high level of TEPs. Prieto et al. (2001) demonstrated a size increase of TEPs with turbulent energy including that produced by zooplankton. Burns at al. (2019) used an oscillating grid devise to demonstrate the abiotic formation of TEPs. These experiments used TEPs to quantify the phase transition from dissolved precursors to particles, but on much larger time scales. Contrary to these published experiments, our short time scales would minimize the impact of any microbes present in the

sample.

Another process that might have helped the aggregation of gels in our filtrations is the drops of filtrate falling from the filter holder into the receiver vessel. The falling drops might have induced the formation of small bubbles that then could have supported the aggregation of gels (Johnson and Cooke, 1980; Robinson et al., 2019). It might be argued that the gels formed spontaneously to arrive at a new dynamic balance between precursors and aggregates after the aggregates had been removed

by filtration (He et al., 2016), but this hypothesis could not explain that the ratio of $POC_2$ $POC_1^{-1}$ was greater than one in samples >1000m depth. We considered the possibility that the environmental conditions on board the ship, such as temperature and pressure promoted the formation of aggregates, but the time between receiving the water sample on board and the first filtration was similar to the period between the first and second filtration, therefore, the sample had been exposed to the on board conditions the same period for both filtrations. One reviewer suggested that inorganic carbon particles might have formed

in deep samples with the pressure change when moved to the surface. Because we did not treat our samples with acid, this inorganic carbon might have led to exaggerated particulate carbon estimates. This process would contradict the finding of high POM in the refiltered samples (POM2) (Figure 6). In conclusion we suggest that the transfer from deep waters to the surface was not responsible for the increase in gels after the first filtration. Biogenic gel formation can be ruled out because the



concentration of microorganisms in the filtrates was much reduced in the filtrates and the time between successive filtration

was too short.

Our interpretation for the high ratio $POM_2\ POM_1^{-1}$ in deep water is that the filtrate has a high potential to form aggregates, but because of the low in situ turbulent energy at this depth, the gels had been prevented to form. If turbulent energy is determining the relative amount of gel aggregates, then the interpretation of deep-sea POM, or the interpretation of hydro-gels to POM ratios is ambiguous because we know little about the in situ turbulent energy distribution and in comparison to the epipelagic,

possibly biogenic turbulence or current boundaries can produce local gel hotspots. Gels influence the sinking rate of particles and the biological pump, the POC/PON ratio and the ecology of prokaryotes associated with particles (Bižić-Ionescu et al., 2018), to better understand these processes we need to know the in situ conditions that are controlling the aggregation of precursors to gels.

If turbulence can produce gel aggregates (gel-HySt), then they might be a higher concentration expected at density

discontinuities where a higher hydraulic shear might exist. MacIntyre et al. (1995) observed higher marine snow concentrations at density discontinuities but they might also be explained by a lower sinking speed. Once gel particles are produced, are they maintained or do they disintegrate to reach a physicochemical equilibrium under low turbulence condition? The existence of a dynamic equilibrium within the size range of gels and gel-precursors (He et al., 2016; Jackson, 2008) is not a new concept, for example Asmala et al. (2014) included in their model a disintegration term for flocculants to maintain a dynamic balance.

Removal of flocculants or gels by filter feeders would be one mechanisms of their removal. Already Baylor and Sutcliffe (1963) showed experimentally that the organic particles produced by bubbling filtered seawater could be used to feed Artemia shrimp. Robinson et al. (2019) documented the use of TEPs produced by bubbling could stimulate bacterial carbon production. How can the dynamic equilibrium between precursor-DOC and gels could have bearings on methodological approaches or ecological concepts? For example: 1) Considering the removal of particles by filter feeders with the primary filtration ($F_1$),

then we have to consider that the turbulence produced by filter feeders induces the concurrent formation of more particles from precursors that can also be consumed by the filter feeder. Consequently, measuring the organic particles before and after a feeding experiment might underestimate the ingestion. 2) With respect to the biological pump, if we consider the participation of gels in the formation of aggregates that are sufficiently big to sink, then this would constitute a loss rate of gels and its precursors at a certain depth. To maintain the dynamic balance new gels could then be formed at that depth, or gels that are

formed in a high turbulence environment at the surface might disintegrate at depth with low turbulence. 3) Because only a fraction of DOM is composed of precursors, the formation of gels and their subsequent transport to other steps constitutes a selective transport of a certain DOM fraction. 4) If gel aggregation is dependent on in situ turbulence, then higher gel concentrations might be found at the depth of frictional layers or high biogenic turbulence (McIntyre et al., 1995).

## 5. Summary and conclusion

The particulate organic material (POM) is a central constituent of the carbon cycle in the ocean because POM drives the biological pump that transports carbon from the surface to the ocean floor where it will be stored for more than 1000 years.





This particle transport is also partially controlling the deoxygenation of mesopelagic (>200m) waters. POM is collected by filtration and consists of sinking aggregates and slow sinking gels. Our results suggest that gels are in rapid dynamic balance with dissolved precursors and the balance depends partially on turbulence, including shear stress during the filtration. The
different shear stress provided by different filtration designs, for example in situ filtration might explain differences in collected POM with different designs. In ecological terms, this suggests that if these particles are removed, for example by filter feeders or by being included into particle aggregates, then dissolved organic precursors could rapidly form new gels to maintain the balance. If the balance between particulate and dissolved organics is partially controlled by in situ turbulence, then any quantification of hydrogels would have to consider the specific conditions in situ. The high concentration of gels found in the
first filtrate of deep water samples suggest that in general bathypelagic gel particles represent a proportionately greater part of particulate organics than at the surface. Apart from biogeochemical consequences our results point to certain aspects of the POM methodology that should receive further attention such as: The method of acidification to remove inorganic carbon, the correction for adsorbed organics and the contribution of bacteria to second filters for correction should be reviewed. Considering that gels can be a very significant part of measured POM, the role of hydrogels on applied processes should be
examined, from the proper interpretation of in situ beam attenuation data, to ocean color remote sensing estimates of POC, to the fouling of membrane-based desalination plants (Bar-Zeev et al., 2009).

**Author contributions**

PVV, CAJ and HM processed samples and wrote the paper.

**Competing interests**

255 The authors declare that they have no conflict of interest

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

**Figure legends**

**Figure 1.** Sequential filtration of seawater samples; sample identifier is indicated.

**Figure 2.** 24 replica pairs of POC and PON were tested for reproducibility of the method. Data pairs include in situ samples,
cultures and refiltered samples. Multiple replicates were graphed with random x or y assignment. The 95% confidence interval of the type 2 regression (dashed lines) and the confidence interval of the population (dotted line) is indicated (SigmaPlot).

**Figure 3.** The effect of vapor acidification on POC and PON using precombusted GFF filters wetted with distilled water. Five experiments using five samples each for acid treatment and blanks. Fresh concentrated HCl was used except where noted. 1.
The desiccator was cleaned with solvent; 2. Desiccator was cleaned with water and detergent; 3. Desiccator cleaned with solvent. The desiccator top was sealed with grease in 5 and 6.

**Figure 4.** The POC (a), PON (b) and POC/PON (c) data from seawater from 950m depth off Ensenada that was originally filtered and then aged for several months. Top filters and bottom filters versus the number of sequential filtration steps (F1 to
F5). In Fig.4d the ratio of lower to upper filter POC is graphed against filtrations steps; the geometric means (bars) are indicated for F1 (0.11) and for F2 (0.17).





**Figure 5.** POC (a), PON (c) and POC/PON (e) ratio of refiltered samples from different depth strata in the Gulf of Mexico and the California Current. The abscissa numbers correspond to the filtration sequence (F1 to F5). Figures b, d, f show the data normalized to the results of the first filtration.

**Figure 6.** Log/log scale (POM2/POM1) ratio versus (depth). Both type 2 regressions are significant ($p<0.05$).

**Figure 7.** Sequential filtration (0.45 m) for TEP samples from different depth ranges: 0-100m: blue; 100-1000m: red; >1000m: black. a) Change in TEP concentration in Gum Xanthan equivalence, b) TEP concentration normalized to the concentration of the first filter. The filtrate used for the 4th filtration was kept during the night in the refrigerator.

**Figure 8.** Bacterial abundance after filtration step i ($B_i$) normalized to prefiltration abundance ($B_i/B_0$) is indicated for filtrations with double GFF filters and single GFF filtrations. In this graph the abscissa numbers indicate the number of passage ($F_i$) through pre-combusted GFF filters before the bacterial abundance was sampled; i.e. position 0 shows the normalized bacterial abundance in the original sample, $i = 2$ indicates the relative abundance after passage through double GFF filters or two separate GFF filtrations. The left ordinate: The numerical approximations for the two sets of data (Eq. 1 and 2). On the right ordinate the relative loss in bacterial abundance (($B_{i-1}- B_i$) $F_{i-1}$ $B_0^{-1}$) was calculated from Eq. 1 and 2 and graphed against $F_i - 0.5$.

**Figure 9.** Measured POC versus the POC calculated to represent the bacterial biomass retained by the GFF filter. Bacterial POC was calculated from bacterial abundance of the sample and a retention efficiency of 50% per GFF passage.





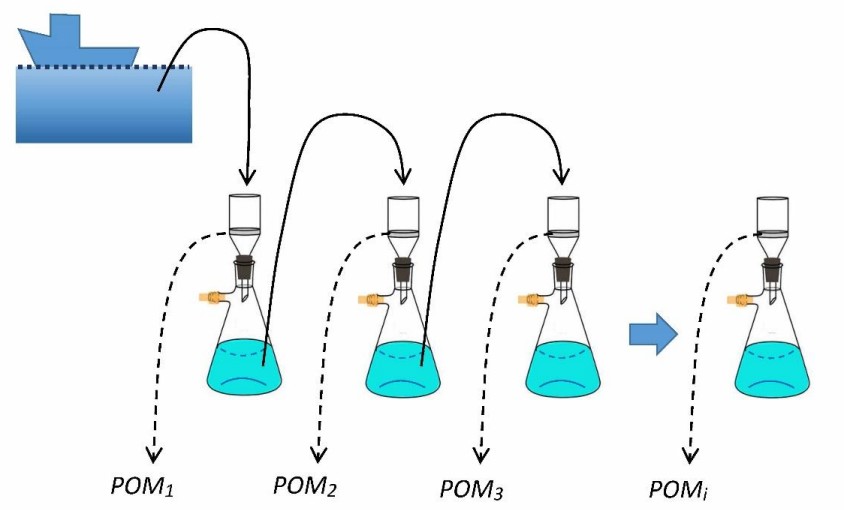

Figure 1. Sequential filtration of seawater samples; sample identifier is indicated.





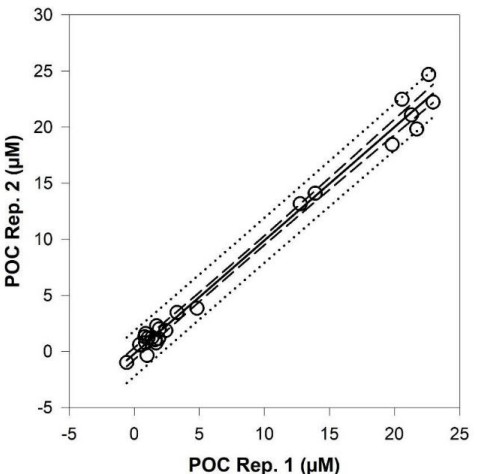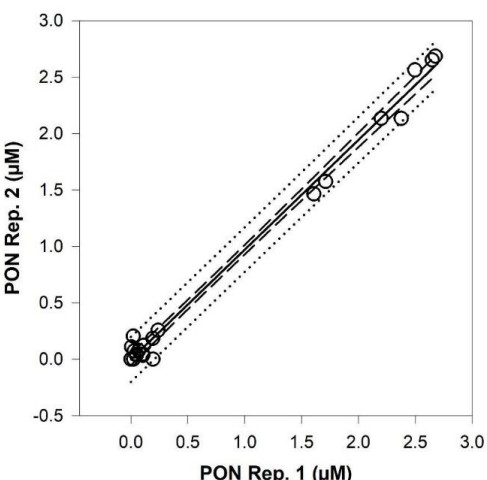

Figure 2. 24 replica pairs of POC and PON were tested for reproducibility of the method. Data pairs include in situ samples, cultures and refiltered samples. Multiple replicates were graphed with random x or y assignment. The 95% confidence interval of the type 2 regression (dashed lines) and the confidence interval of the population (dotted line) is indicated (SigmaPlot).





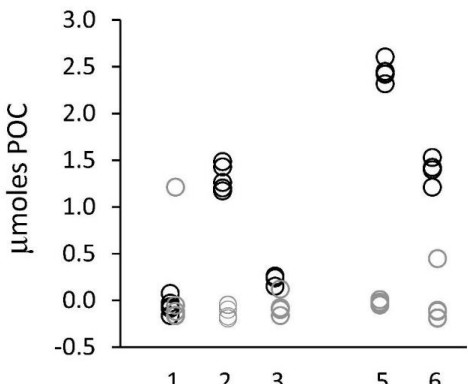
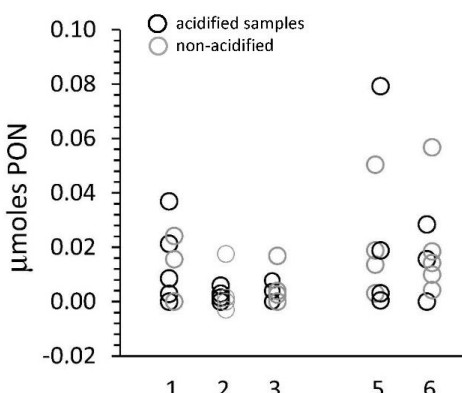

Figure 3. The effect of vapor acidification on POC and PON using precombusted GFF filters wetted with distilled water. Five experiments using five samples each for acid treatment and blanks. Fresh concentrated HCl was used except where noted. 1. The desiccator was cleaned with solvent; 2. Desiccator was cleaned with water and detergent; 3. Desiccator cleaned with solvent. The desiccator top was sealed with grease in 5 and 6.



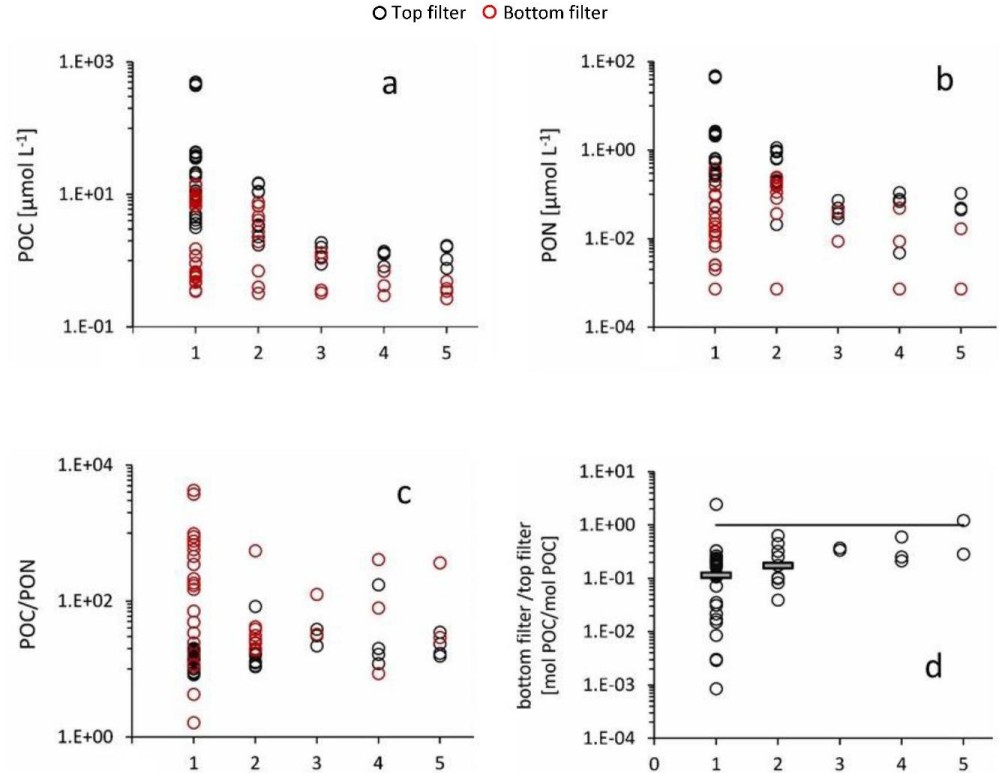

Figure 4. The POC (a), PON (b) and POC/PON (c) data from seawater from 950m depth off Ensenada that was originally filtered and then aged for several months. Top filters and bottom filters versus the number of sequential filtration steps ($F_1$ to $F_5$). In Fig.4d the ratio of lower to upper filter POC is graphed against filtrations steps; the geometric means (bars) are indicated for $F_1$ (0.11) and for $F_2$ (0.17).




Figure 5. POC (a), PON (c) and POC/PON (e) ratio of refiltered samples from different depth strata in the Gulf of Mexico and the California Current. The abscissa numbers correspond to the filtration sequence ($F_1$ to $F_5$). Figures b, d, f show the data normalized to the results of the first filtration.





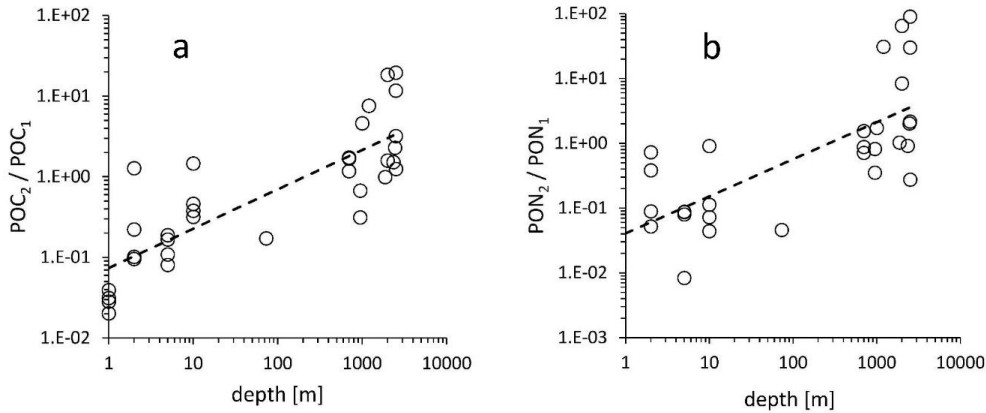

Figure 6. Log/log scale ($POM_2/POM_1$) ratio versus (depth). Both type 2 regressions are significant ($p<0.05$).



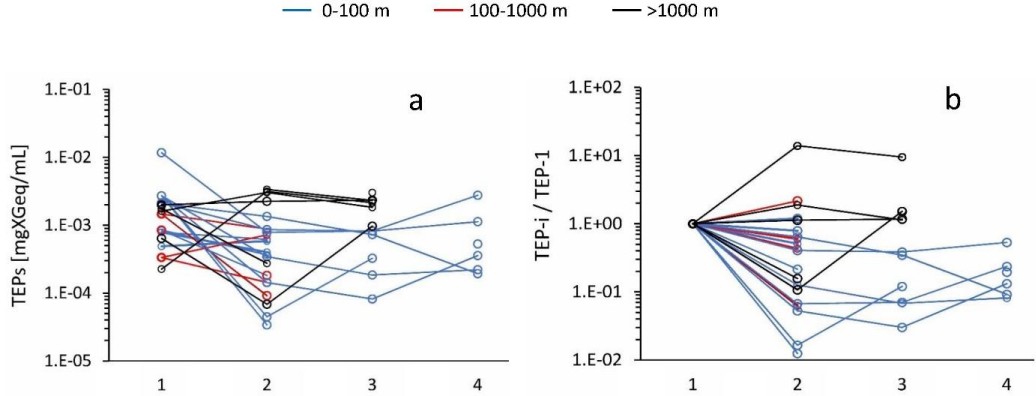

Figure 7. Sequential filtration (0.45 m) for TEP samples from different depth ranges: 0-100m: blue; 100-1000m: red; >1000m: black. a) Change in TEP concentration in Gum Xanthan equivalence, b) TEP concentration normalized to the concentration of the first filter. The filtrate used for the 4th filtration was kept during the night in the refrigerator.



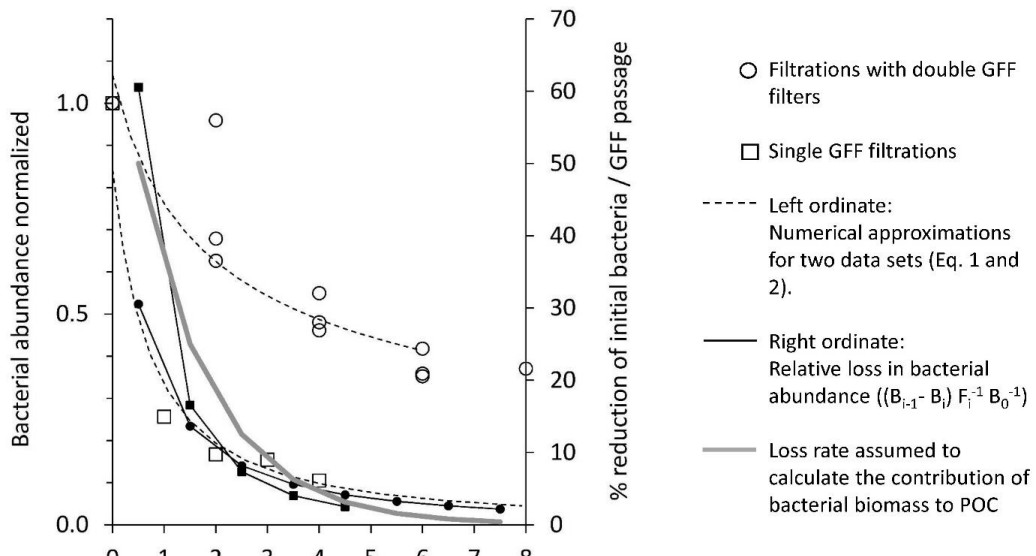

Figure 8. Bacterial abundance after filtration step $i$ ($B_i$) normalized to prefiltration abundance ($B_i/B_0$) is indicated for filtrations with double GFF filters and single GFF filtrations. In this graph the abscissa numbers indicate the number of passage ($F_i$) through pre-combusted GFF filters before the bacterial abundance was sampled; i.e. position 0 shows the normalized bacterial abundance in the original sample, $i = 2$ indicates the relative abundance after passage through double GFF filters or two separate GFF filtrations. The left ordinate: The numerical approximations for the two sets of data (Eq. 1 and 2). On the right ordinate the relative loss in bacterial abundance ($(B_{i-1} - B_i) F_i^{-1} B_0^{-1}$) was calculated from Eq. 1 and 2 and graphed against $F_i$ - 0.5.





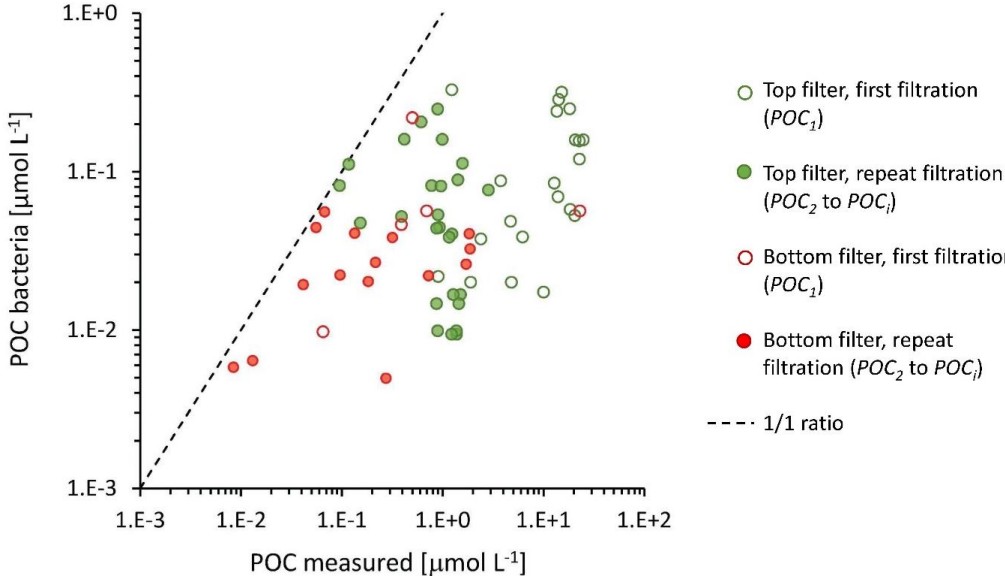

Figure 9. Measured POC versus the POC calculated to represent the bacterial biomass retained by the GFF filter.
Bacterial POC was calculated from bacterial abundance of the sample and a retention efficiency of 50% per GFF
passage.