# Peer review of "Rapid abiotic transformation of marine dissolved organic material to particulate organic material in surface and deep waters."

_Biogeosciences, 2020_

## Referee Comment (RC1) · Anonymous Referee #1 · 16 Sep 2020

In this study, seawater samples were repeatedly filtered, and the POC concentrations in re-filtered samples were assessed. It is important to assess the validity of established technique (here, filtration with GF/F filter), but this manuscript is too immature. The most serious concern is the validity of the discussion on the abiotic aggregation during filtration procedure. The authors considered that detected POC in re-filtered sample is derived from aggregated POC during filtration procedure. However, they did not provide any clear evidence for aggregation. If aggregation occurs within short period as the authors considered, conversion from DOM to POM might easily occur under turbulence in natural seawater as the authors discussed, but I feel suspicious. Many studies had previously shown aggregation process by shear stress. I think timescale

of those experiments would be several tens minutes to hours. In this study, they said aggregation occurred within 4 seconds, which is quite short. If the authors provide clear evidence, this finding could have implication to change the concept of DOM-POM continuum. I think there is no clear evidence to support the aggregation in such short period. Simply, the particles passing through 1st filter might be trapped in the second filter. As the authors mentioned, structure of GF/F filter would be not uniform. On the contrary to membrane filter, cut-off by GF/F filter would be variable. Therefore, it is probable that a part of particles with size around 0.7 $\mu$m is sometimes trapped but sometimes not. In addition, I feel English is sometimes awkward, so the manuscript should be checked by professional correction service.

Introduction The strategy of this study is quite obscure. The authors should clearly state the aim of measurement of bacteria and TEP in introduction section. I cannot clearly understand what is new in this study. They should add more reference.

Line 26: I think the cut-off size is generally 0.7 $\mu$m. On the HP in Millipore, the pore size is described as 0.7 $\mu$m.

Line 30: I think the term "membrane enclosed particles" is not generally used. The authors did not apply any specific method the particle with and without membrane. So, I believe this term is not appropriate.

Line 34-36: Details of CHN instrument is not important in this study. I recommend deletion.

Line 39-42: Please describe how to treat the sample blank in the previous studies with more detail, because it relates with this study. Were the methods used in this study same with the previous study?

Line 45-47: From this sentence, readers will think that the particles are same regardless the name was different. However, the meanings of the words; EPS, SAG, mucilage, perfect slime, TEP, CSP, were overlapped, but not same. For example, TEPs

are alcian-blue stainable particles, but CSP are coomassie stainable particles. Please clarify.

Line 47-49: I cannot understand why the authors mention that they do not consider marine snow in this sentence. I recommend deletion.

Line 57-58: Again, I think the authors should not separately describe MEPs and other particles, because this study did not distinguish the particles based on with/without membrane.

Line 60-62: In my understanding, for TEP measurement, filter with pore size of 0.4 $\mu$m has been used.

Line 71-74: In the two sentences in this section, the first one introduces aggregation of DOC in surface seawater, but second one is the description about deep sea. How did the author consider the difference of the samples? The ratios of POC/DOC would be considerably different across the depth, so it should be carefully compared.

Line 77-85: The author did not cite any reference in this section. For example, is there any literature about the relative contribution of hydrogels to the POC across the depth? They mentioned less change in hydrogel with depth, but please provide literature. In addition, they considered prokaryote cell passing the first filter is partly retained on the second filter, but is it not just absorption? How do they think the difference of retainment and absorption?

Line 88-89: Please provide reference.

Line 92-102: No literature was introduced this section. They need to carefully provide literature (e.g., termination of aggregation in phytoplankton bloom, acceleration of aggregation with turbulence).

Methods Line 114-119: I think the pressure at 5 kPa is relatively lower than standard protocol. They mentioned it took 1 hour to complete filtration for each filtration step. I feel it is slightly longer. Aggregation would time-dependently occur. If aggregation is

important as authors consider, filtration with long period could be an artifact to induce aggregation. How much volume did they filter? In addition, what is Fi? In Fig. 1, POM1, 2 and 3 plus POMi were shown. I understand POM1-3 as sequential filtration. In the figure, I cannot understand what kind of sample is introduced to filtration funnel.

Line 122-137: The information about measurement of POC/PON would be redundant. I recommend more concise. On the other hand, I would like to recommend addition of detection limit and limit of quantification. Probably, the concentration of POC on the second filter would be low, so the detection limit would affect the quality of the data.

Line 158-159: Please provide reference for the value of TEP carbon to XG.

Line 161-165: How long were the samples stored before counting? Please clarify.

Results The quality of figures is bad. Lines and plots are sometimes overlapped.

Line 175-176: I would like to confirm how to treat the sample in the accident. Two samples were measured together, and the values were provided as a half of the measurement, right? If so, I strongly recommend to deletion the data.

Line177-184: When I read these sentences, I thought no difference in the treatment for experiment 1-3. Thereafter, I found difference among 1-3 in legend of Fig. 3. This is quite tricky, so please rewrite.

Line 191-199: First, the presentation of data using Fig. 4 is not appropriate. The log-scale is not useful to compare the data of top and bottom filter, because some plots were overlapped. I recommend boxplot. Difference of POC and PON between top and bottom filter should be statistically analyzed. The increase in the POC/PON ratio with filtration step should be also statistically analyzed. For the data analysis, I think the detection limit would be important. I suspect that POC on some of the bottom filter is under detection limit. Please clarify.

Line 199-203: I think it is difficult to discuss the aggregation due to shear stress based on the increase in bottom/top ratio. For example, if the shape of particles is ellipse,

the efficiency of filtration would be not 100%. Natural particles would have complicate shape, so the particles with the size around cut-off of GF/F would sometimes pass, but sometimes retained. After sequence of filtration, most of such particles would be removed.

Line 204-207: The Fig. 5 is bad. Most of symbols and lines are overlapped. So, it is quite difficult to check the datasets.

Line 207-219: The authors showed only geometric means, but the error should be also provided. This study focuses on methodological aspect, so the assessment of confidence is essential.

Line 225-226: What is the "expected range for filters of 0.45 $\mu$m"?

Line 226-236: Fig. 7 is also bad quality. The data seems to be highly variable among replication. The author should interpret the data with consideration on technical error.

Line 229: What kind of statistical method was applied?

Line 238-239: These two sentences should be moved to introduction.

Line 224-257: Regarding TEP and bacterial cell number, no information of the concentration was provided. The raw data of the concentration is important to assess the validity of the methods.

Line 246-257: This section should be moved to interpretation.

Interpretation Line 259-262: The reason why the authors considered that aggregation due to fluid shear contribute to retainment of POC. If the reason is only increase in top/bottom ratio, I cannot agree as pointed out in the comment for Line 199-203. Please discuss with more clear evidence. The author should compare with the past study on aggregation under fluid shear. In Drapeau et al. (1994), aggregation under laminar shear flow was observed along time course using Couette device. They showed aggregation, but the timescale is several tens minutes to hours. Aggregation

within 4 seconds is quite short. So, if the authors want to say, they need to provide more prominent evidence.

Line 272-274: Regarding aggregation by bubbling, a report had been recently published, in which the timescale to form aggregation is also provided. Wada et al. (2020) Journal of Oceanography, 76, 317-326

Line 282-290: Retainment of bacteria would not be evidence that new hydrogels appeared. There are a variety of organic particles other than bacterial cells, and behavior of those other particles would be different from that of bacteria. In addition, size of bacterial is highly variable. Since data of bacterial abundance was provided as ratios not concentration as pointed out by the comment for Line 224-257, it is impossible to justify whether this study can be generalized.

Line 307-310: Treatment with acid would remove not only inorganic particles. DIC in seawater sample would be absorbed to filter, and acid remove them. The concentration of DIC is generally 1-2 order of magnitude higher than POC and DOC, so it could lead overestimation.

Line 312-326: Here, the authors compared the present study with the previous ones. Such comparison should be done in introduction section, because when I read introduction, I felt the novelty of this study is unclear.

Line 330-334: I think hydrogels would also partly pass through the first filter as well as bacterial cells. I cannot understand why the author only mention aggregation regarding hydrogel.

Line 335-337: Please describe more detail of the literature; Maske and Garcia (1994). How much amount of carbon was absorbed? Is it possible to explain the results in this study from the amount of carbon absorbed in the literature? Was the seawater used in the literature comparable with the present study?

Line 339-341: The connection between this and past sentence is unclear. What is the

reason why the author provided the assumption?

Line 343-350: The possibility that the authors proposed is not limited to bacteria. I feel strange why they put this information just before the discussion about bacteria.

Line 359: What does "this assumption" imply here? Please clarify.

Line 363-371: I also think bacterial cell trapped on 2nd filtration would be small, and carbon content per one cell could be low. Therefore, estimate of carbon in bacterial cells in Fig. 9 would be not reliable, and the discussion based on Fig. 9 is meaningless.

Line 372-378: Again, less contribution of bacteria is not evidence of aggregation. The authors should provide evidence.

Line 385-386: If shear stress during filtration really affect the concentration of TEP, as the author described, interpretation of TEP is difficult. However, the authors did not show clear evidence in this study.

Line 400-401: The difference in staining between Gum Xanthan and natural gels is not problem, because Gum Xanthan is just a standard compound. Problem is the ratio of TEP-carbon to Gum Xanthan equivalent is variable among samples.

Line 405-416: I cannot follow the authors' thought in that they described rapid and non-biogenic phase transition is supported. What is the direct evidence for this finding? In the section from 405-415, they introduce the previous reports, but suddenly said "Our data support ∼". Here, what is the "our data"? POC, TEP or bacteria? Please explain step by step.

Line 420-424: The proposed idea of the authors is not sufficiently proved.

Line 441-450: Many uncontrolled factors are considered in this section. If the condition on board has serious effect on the data, this study has serious problem in the viewpoint of reproducibility, because no one can treat sample with same manner in this study.

Line 444: I think reviewers' comment should not be directly mentioned in the main text.

Line 451-458: The authors should carefully interpret the results. Aggregation should be supported by clear evidence. Throughout this manuscript, I cannot find any clear evidence for aggregation.

Line 459-479: As the author mentioned, effect of turbulence on biogeochemical cycle has been recently attracted phenomenon. However, in this study, they did not provide any information on turbulence or shear stress. In addition, I think aggregation in quite short period is suspicious.

---

## Referee Comment (RC2) · Anonymous Referee #2 · 20 Sep 2020

The paper by Villaverde et al. presents the results from testing the method for measuring particulate organic matter (POM) in seawater samples, focusing on artifacts associated with filtration. They discuss the implications of these measurements, emphasizing their relevance for conversion of dissolved organic matter to particulate by coagulation processes.

The paper presents interesting new ideas that are worth publication, but the presentation needs to be improved. There is a lot of repetition of some of the ideas that can make reading this numbing.

Issues:

Use of the expression "membrane enclosed particles (MEPs)" is a rather peculiar way to refer to non-TEP, non-gel particles, given that it is mixture of fecal pellets, diatom frustules, dead algae, dead animals, dust, ..., as well as bacteria and algal cells.

Logan (Logan 1993 L&O 38: 372; Logan et al. 1994. L&O 39: 390) has made similar observations on the collection by glass fiber filters of organisms that are smaller than the ostensible pore sizes. He used classical filtration theory in his analysis. Similar processes should be occurring with the filtration of colloidal organic matter. That is, the actual mechanism for collection may not be the production of larger particles passing through the filters but be related to direct filtration processes on the colloid removal.

It would be nice to see what fraction of the "dissolved" gels is removed by each pass through the filters. This would involve providing filtered volumes and DOM concentrations.

Need to have clear separation between Methods, Results, and Interpretation (also known as Discussion) sections. For example, - - L172-174 and L185-190 belong in the Methods section. - - L231-236, L238-239, L248-258 all include comparisons of results with previous literature and belong in the Discussion section.

The Interpretation/Discussion section needs to be condensed. Given the relatively few experimental findings and small amount of data interpretation, having 7 $\frac{1}{2}$ pages for the discussion of a 15 page manuscript is a lot.

Stylistic suggestions: 1. Use clear demarcation of new paragraphs. For example, indent the beginning of each paragraph or have a blank line between paragraphs. 2. Use different symbol shapes (+, x, o...) for different data sets rather than the same symbols and different colors. The colors are hard to differentiate, particularly for copies made on black and white printers. 3. Break the really long sentences into more than one. Reading technical papers is difficult enough without having to tease apart complicated sentence structures to understand the arguments. 4. Make the notation consistent throughout the manuscript. For example, L13 has POM2/POMi. Should this

not be POM2/POM1? In L115-6, POM1, POC1 and PON1 should be italicized and the 1s should be subscripted. Place a space between the number and the unit when giving data (e.g. L13, 14, 26...).

Please also note the supplement to this comment:
https://bg.copernicus.org/preprints/bg-2020-291/bg-2020-291-RC2-supplement.zip

---

## Referee Comment (RC3) · Anonymous Referee #3 · 21 Sep 2020

This study presented a phenomenon of POM formation in filtered DOM filtrates. Dynamic exchanges between DOM and POM are well-known and can be caused by an array of physical, chemical, and biological factors, including sunlight, pH, cations, phytoplankton, hydrodynamics, etc. It's also related to concentration, composition, and size of DOM and POM. It's an interesting but complicated topic. However, it's unclear in this study what're the detailed organic matter composition and operation conditions of filtration. According to the description in "Methods" section, it seemed that the filtration procedures have been partly done onboard. Photo-flocculation can occur but the sunlight doses onboard and in the lab are supposed to different. Moreover, why did the authors use distilled water instead of ultrapure MQ-H2O during the experiments?

[Figure]

Distilled water itself may contain some amount of organic matter. In addition, the authors need to take into account of adsorption and subsequent desorption of retained colloidal DOM. Also, using a second filter to estimate filtration blank is problematic.

Below are some specific comments: Throughout the manuscript: 1)add a space between number and unit; 2) change GFF filters to GF/F; 3) subscript numbering of filtration times; 4) pay attention to the punctuation and grammars.

Line 19 change "Organic particulate matter (POM)" to "Particulate organic matter"

Lines 86-88 This sentence is obscure and need to be re-phrased.

Lines 101-102 This is speculative. Or please provide reference.

Lines 109-112 Algal cultures are prone to form aggregates within a period of time. Please take this into account.

Lines 114 and 122 Correct the degree symbol

Line 137 "reproducibility o deep...", do you mean "reproducibility of deep..."

Line 157 0.45 mm or 0.45 um?

Lines 157-158 What are the grades and brands of these chemicals

Line 162 Superscript "-1" in the unit

Line 311 Change "4.1.3" to "4.1.2" and also the following subsection numbers

Figure 3 "precombusted" or "pre-combusted"? Please be consistent through the manuscript

Figure 7 missing unit for the filter pore size

---

## Author Comment (AC1) · 30 Oct 2020

Anonymous Referee #1 In this study, seawater samples were repeatedly filtered, and the POC concentrations in re-filtered samples were assessed. It is important to assess the validity of established technique (here, filtration with GF/F filter), but this manuscript is too immature. The most serious concern is the validity of the discussion on the abiotic aggregation during filtration procedure. The authors considered that detected POC in re-filtered sample is derived from aggregated POC during filtration procedure. However, they did not provide any clear evidence for aggregation. If aggregation occurs within short period as the authors considered, conversion from DOM to POM might easily occur under turbulence in natural seawater as the authors discussed, but I feel suspicious. Many studies had previously shown aggregation process by shear stress. I think timescale of those experiments would be several tens minutes to hours. In this study, they said aggregation occurred within 4 seconds, which is quite short. If the authors provide clear evidence, this finding could have implication to change the concept of DOM-POM continuum. I think there is no clear evidence to support the aggregation in such short period. Simply, the particles passing through 1st filter might be trapped in the second filter. As the authors mentioned, structure of GF/F filter would be not uniform. On the contrary to membrane filter, cut-off by GF/F filter would be variable. Therefore, it is probable that a part of particles with size around 0.7 $\mu$m is sometimes trapped but sometimes not. In addition, I feel English is sometimes awkward, so the manuscript should be checked by professional correction service. Introduction The strategy of this study is quite obscure. The authors should clearly state the aim of measurement of bacteria and TEP in introduction section. I cannot clearly understand what is new in this study. They should add more reference.

GENERAL RESPONSE: The reviewers were understandably skeptical of the process of aggregation of dissolved organics promoted by hydraulic stress on the time scale of seconds. To answer doubts we recently did some additional experiments where we compare pre-filtered coastal surface water that was directly re-filtered as in the data reported in the original manuscript (non-stressed) or passed through a capillary (0.5mm ID) and then re-filtered (stressed). In the new Fig. 4 we show the difference between stressed and non-stressed POC and PON. We show that the difference between stressed minus non-stressed POC and PON are significantly higher than zero. It is difficult to compare quantitatively the hydraulic stress exposure of passing through a capillary or through a filter, but the time scales of exposure are similar. The GF/F filters of stressed and non-stressed samples will contain bacterial biomass, adsorbed organics and gels formed during the prefiltration, but the difference between both samples should be due to aggregated dissolved organics formed by passage through the

capillary. See details of our simple and easily reproducible experiment in methods and results. As suggested we calculated a lower limit of detection for the POC and PON method and eliminate the data below this limit from figures and interpretation. One exception are the data were we compared the effect of sample acidification. We had included TEP data in the original manuscript submitted because we wanted to make a link to the abundant TEP literature. We eliminated the TEP data from the new manuscript because they did not present a pattern with statistical significance. We left some of TEP related discussion. Hopefully without the TEP data the new manuscript is more concise. The old Fig. 4 (POC and PON with refiltrations) was eliminated, because some of that information is reported in Fig. 5 and to make the manuscript more compact. As suggested by reviewers we changed the Fig. 5 to distinguish the patterns better even when printed in black and white. We added a figure 9 with a conceptual sketch.

SPECIFIC RESPONSE: Line 26: I think the cut-off size is generally 0.7 $\mu$m. On the HP in Millipore, the pore size is described as 0.7 $\mu$m. Response: Corrected

Line 30: I think the term "membrane enclosed particles" is not generally used. The authors did not apply any specific method the particle with and without membrane. So, I believe this term is not appropriate. Added explanation why MEPs are defined: '…..defined here as membrane enclosed particles (MEP) to indicate that they should resist deformation during filtration more than gel-like aggregations of organics.'

Line 34-36: Details of CHN instrument is not important in this study. I recommend deletion. Response: This sentence about CHN method was included because it might have relevance for possible methodological artifacts.

Line 39-42: Please describe how to treat the sample blank in the previous studies with more detail, because it relates with this study. Were the methods used in this study same with the previous study? Response: The sentence was changed to include more details: '… . . . using different approaches. Moran et al (1999) used the intercept of POC

when plotting POC against volume of sample. Because the intercept was higher than the sample blank (dry GFF filter) they considered the difference to be due to adsorbed dissolved organics, assuming that the adsorbed organics would be independent of the volume filtered. Turnewitsch et al. (2007) also used the intercept method but they observed nonlinear behavior contrary to the original concept. Liu et al. (2005) used a second filter underneath the sample filter assuming that both filters in the stack adsorbed the same amount of DOC. Rasse et al. (2017) compared intercept and second filter blanks and settled on intercept blanks because they produced lower and more similar values in two cruises. Novak et al. (2018) looked at the potential of dissolved organic adsorption by prefiltering samples with filters of smaller pore size than GF/F filters and they found that the retained organics on the filters increased with the volume of the prefiltered sample. Despite the different approaches to estimate and correct for adsorbed dissolved organics on GF/F filters there is no generally accepted method to correct for these non-particulate organics.

Line 45-47: From this sentence, readers will think that the particles are same regardless the name was different. However, the meanings of the words; EPS, SAG, mucilage, perfect slime, TEP, CSP, were overlapped, but not same. For example, TEPs are alcian-blue stainable particles, but CSP are coomassie stainable particles. Please clarify. Response, changed to: 'Hydrogels have many different names in the literature such as exopolymeric substances (EPS), self-assembling gels (SAG), mucilage, the perfect slime (Flemming et al., 2017). Specific chemical fractions of hydrogels are defined by their staining method such as transparent exopolymer particles (TEPs, Passow and Alldregde, 1995) and Coomassie stainable particles (CSP; Long and Azam, 1996).'

Line 47-49: I cannot understand why the authors mention that they do not consider marine snow in this sentence. I recommend deletion. Response: Deleted Line 57-58: Again, I think the authors should not separately describe MEPs and other particles, because this study did not distinguish the particles based on with/without membrane.

Response: Deleted

Line 60-62: In my understanding, for TEP measurement, filter with pore size of 0.4 $\mu$m has been used. Response, correct. The TEP experimental part and data were eliminated in the edited manuscript.

Line 71-74: In the two sentences in this section, the first one introduces aggregation of DOC in surface seawater, but second one is the description about deep sea. How did the author consider the difference of the samples? The ratios of POC/DOC would be considerably different across the depth, so it should be carefully compared. Response, changed to 'If we assume for the deep ocean DOC to be about 50 ïA■mol/L and POC to be about 1 ïA■mol/L, and that the ratio of DOC to gels is the same throughout the water column, then we expect that more carbon is stored in gels than in POC retained by filters.'

Line 77-85: The author did not cite any reference in this section. For example, is there any literature about the relative contribution of hydrogels to the POC across the depth? They mentioned less change in hydrogel with depth, but please provide literature. In addition, they considered prokaryote cell passing the first filter is partly retained on the second filter, but is it not just absorption? How do they think the difference of retainment and absorption? Response: We rewrote the paragraph: 'POM also includes part of the prokaryotes biomass. Because part of the marine prokaryotes community is smaller than the cut-off size of precombusted GFF filters they form part of the filtrate. The relative proportion of prokaryote biomass included in POM will depend on the size distribution of the prokaryotes, and therefore on their physiological status. The prokaryote cells that passed the GFF filter will partially be retained when a second GFF filter is placed below the top filter. The POC of the lower filter is normally interpreted as adsorbed dissolved organics but the organics measured on this second filter will include retained prokaryote cells that had passed the first filter, as our data below show. We propose that another organic component retained by the second filter is made up of hydrogels, these gels were formed from DOC that passed the first GFF

filter. We expect that the aggregates formed during the passage through the first GF/F filter are of sufficient size to be captured by the second filter. The relative contribution of these hydrogels to the measured POC depends on the water depth, because the ratio of DOC to MEPs increases with depth. We assume that the POM retained by the GFF filter always includes gels and that in a second filtration the gels forma a significant part of the retained organics.'

Line 88-89: Please provide reference. Response: Added

Line 92-102: No literature was introduced this section. They need to carefully provide literature (e.g., termination of aggregation in phytoplankton bloom, acceleration of aggregation with turbulence). Response: Added comment

Methods Line 114-119: I think the pressure at 5 kPa is relatively lower than standard protocol. They mentioned it took 1 hour to complete filtration for each filtration step. I feel it is slightly longer. Aggregation would time-dependently occur. If aggregation is important as authors consider, filtration with long period could be an artifact to induce aggregation. How much volume did they filter? In addition, what is Fi? In Fig. 1, POM1, 2 and 3 plus POMi were shown. I understand POM1-3 as sequential filtration. In the figure, I cannot understand what kind of sample is introduced to filtration funnel.

Response and changes in this paragraph: Changed: 'We used sequential vacuum filtration (Fig. 1) with careful control of differential pressure (5 kPa) which is lower than most POM protocols stipulate) to avoid forcing gels through the filter. We did compare this low pressure with higher differential pressure and sometimes found less POC was retained above 0.02 MPa (data not shown).' Response: Filtrations of oceanographic samples took about one hour and only afterwards, the filtrate could be refiltered. The time period necessary for the first filtration depended on organic load and volume filtered, but we tried to keep filtrations to about one hour, which resulted in different volumes that were filtered. Response: The reviewer is correct that some of the subscript numbers that indicated the filtration step in the sequence were not printed out correctly

as subscripts. In Fig. 1 the solid arrows from POM1 to POM2, etc. are supposed to indicate the use of the filtrate of POM1 as sample for POM2. We added to the legend of Fig.1 'The solid arrow pointing from the filtrate of POM1 to the filter funnel of POM2 is supposed to indicate that the filtrate was refiltered.'

Line 122-137: The information about measurement of POC/PON would be redundant. I recommend more concise. On the other hand, I would like to recommend addition of detection limit and limit of quantification. Probably, the concentration of POC on the second filter would be low, so the detection limit would affect the quality of the data. Response: Added: We calculated the lower limit of detection based on our use of dry GF/F filters as baseline, that concentrations were calculated from the difference of signal areas of sample and dry GF/F, and that the lowest sample area would show a signal equal to a dry GF/F filter. If ïA̧şF is the standard deviation of the dry GF/F peak areas measured during instrument runs, and S [micromol/area] is the slope of the standard regression, and the lower limit of detection (LLD, micromol) is three times the standard deviation of the baseline, then LLD = 2 S sigma( 2 sigmaˆ2) Eq.1 With equation 1 an average LLD can be calculated using data from different instrument runs, arriving at a LLD for POC (LLDPOC: 0.4 micromol) and for PON (LLDPON: 0.02 micromol). We eliminated from the data set the data below these limits, except for Figure 2 where we compare blank filters with and without prior acid treatment. Equation 1 does not account for noise in the data due to filtration and sample inhomogeneity's.

Line 158-159: Please provide reference for the value of TEP carbon to XG. Response: TEP data were excluded from the new manuscript

Line 161-165: How long were the samples stored before counting? Please clarify. Response: Added: 'Within a week the sample was incubated. . . . . . . . .. . .'

Results The quality of figures is bad. Lines and plots are sometimes overlapped. Response: Figure 5 was changed in style and a few data points were eliminated because they were below the lower limit of detection. Hopefully the figure is now easier to read.

Line 175-176: I would like to confirm how to treat the sample in the accident. Two samples were measured together, and the values were provided as a half of the measurement, right? If so, I strongly recommend to deletion the data. Response: The two filters processed together were replicates. We marked this sample with a square symbol to identify it. Because the concentration is close to zero, both samples had to be close to zero. If the concentration would have been higher, there might have been doubts about the similarity of concentrations of either replicate. See also below.

Line177-184: When I read these sentences, I thought no difference in the treatment for experiment 1-3. Thereafter, I found difference among 1-3 in legend of Fig. 3. This is quite tricky, so please rewrite. Response, corrected: 'In experiment #3 (Fig. 3) two acid treated filters were accidentally joined in measurement, here we report only one value of half the POC and PON measured and the results are marked with squares. In experiment #5 the HCl was more than 0.5 years old. Experiment #6 used new HCl. The results in Fig. 3 were from separate experiments performed over a period of more than half a year. In Figure 3 the POC in experiments #2, 3, 5 and 6 , yielded significant differences (p<0.05) between acidified (A) and non-acid exposed (B) samples. POC #1 showed no significant difference between A and B samples (p>0.05). The average POC values were: #1: A= -0.05 ïA▪moles, B= 0.14 ïA▪moles; #2: A= 1.31, B = -11 ïA▪moles; #3: A= 0.23, B= -0.06 ïA▪moles; #5: A= 2.44, B= -0.02 ïA▪moles; #6: A= 1.39, B= -0.03 ïA▪moles. Four of the no-acidified samples showed negative POC values. The average POC of all non-acidified samples was -0.017 ïA▪moles, i.e. the distilled water wetted GF/F samples had on average lower values than the dry GF/F used to define the zero. The PON data in Figure 3 showed no significant differences (p>0.05) between acidified (A) and non-acid exposed (B) samples.'

Line 191-199: First, the presentation of data using Fig. 4 is not appropriate. The logscale is not useful to compare the data of top and bottom filter, because some plots were overlapped. I recommend boxplot. Difference of POC and PON between top and bottom filter should be statistically analyzed. The increase in the POC/PON ratio with

filtration step should be also statistically analyzed. For the data analysis, I think the detection limit would be important. I suspend that POC on some of the bottom filter is under detection limit. Please clarify.

Response: Yes, old Fig.4 needed improvement in concept and form. Because the data in old Fig. 4 were obtained from different sample the interpretation of the data is more difficult. We included this figure initially because we wanted to show that even after double filter filtration the GF/F filter still could retain organics, and show that it took up to four filters to reach a steady concentration. One interpretation of this pattern would be the formation of gel by hydraulic stress. We eliminated the old Fig.4 partially because some of the refiltration data are also shown in Fig. 5, although not through double filters. With the new Figure 4 we can show more directly the effect of hydraulic stress and we therefore decided to substitute the old with the new Fig.4.

Line 199-203: I think it is difficult to discuss the aggregation due to shear stress based on the increase in bottom/top ratio. For example, if the shape of particles is ellipse, the efficiency of filtration would be not 100%. Natural particles would have complicate shape, so the particles with the size around cut-off of GF/F would sometimes pass, but sometimes retained. After sequence of filtration, most of such particles would be removed. Response: See Fig. 8 of bacteria passing through sequential GF/F filters and our discussion.

Line 204-207: The Fig. 5 is bad. Most of symbols and lines are overlapped. So, it is quite difficult to check the datasets. Response: Please see changed figure.

Line 207-219: The authors showed only geometric means, but the error should be also provided. This study focuses on methodological aspect, so the assessment of confidence is essential. Response: The change in POM2/POM1 ratio with depth is shown in Figure 6 and the statistical data included. The information in old manuscript Line 207-219 were changed and shortened.

Line 225-226: What is the "expected range for filters of 0.45 $\mu$m"? Response: TEP

data were excluded from the new manuscript

Line 226-236: Fig. 7 is also bad quality. The data seems to be highly variable among replication. The author should interpret the data with consideration on technical error. Response: TEP data were excluded from the new manuscript

Line 229: What kind of statistical method was applied? Response: TEP data were excluded from the new manuscript

Line 238-239: These two sentences should be moved to introduction. Response: Moved

Line 224-257: Regarding TEP and bacterial cell number, no information of the concentration was provided. The raw data of the concentration is important to assess the validity of the methods. Response: TEP data were excluded from the new manuscript

Line 246-257: This section should be moved to interpretation. Response: Partially moved

Interpretation Line 259-262: The reason why the authors considered that aggregation due to fluid shear contribute to retainment of POC. If the reason is only increase in top/bottom ratio, I cannot agree as pointed out in the comment for Line 199-203. Please discuss with more clear evidence. The author should compare with the past study on aggregation under fluid shear. In Drapeau et al. (1994), aggregation under laminar shear flow was observed along time course using Couette device. They showed aggregation, but the timescale is several tens minutes to hours. Aggregation within 4 seconds is quite short. So, if the authors want to say, they need to provide more prominent evidence. Response: In the newly added data of prefiltered seawater passing through a capillary, the exposure to hydraulic stress was less than 5 s. The last part of Interpretation mentioned different publications that showed the experimental aggregation of organics through turbulence. All these reported aggregation on much larger timescales. We added 2 related references.

Line 272-274: Regarding aggregation by bubbling, a report had been recently published, in which the timescale to form aggregation is also provided. Wada et al. (2020) Journal of Oceanography, 76, 317-326 Response: Thank you, helpful reference!

Line 282-290: Retainment of bacteria would not be evidence that new hydrogels appeared. There are a variety of organic particles other than bacterial cells, and behavior of those other particles would be different from that of bacteria. In addition, size of bacterial is highly variable. Since data of bacterial abundance was provided as ratios not concentration as pointed out by the comment for Line 224-257, it is impossible to justify whether this study can be generalized. Response: We partial removal of bacteria as the most obvious explanation for POM in filtrates in surface waters. In deep water samples where the filtrate has more POM that the original filter retained there is no question that POM was formed from DOM. Hopefully the capillary experiments help to make our interpretation more acceptable.

Line 307-310: Treatment with acid would remove not only inorganic particles. DIC in seawater sample would be absorbed to filter, and acid remove them. The concentration of DIC is generally 1-2 order of magnitude higher than POC and DOC, so it could lead overestimation. Response: In retrospect, I would argue that with proper precaution (no greased desiccator seals) we could have acidified the samples; our non-acidified samples should be valid because we interpret differences between filters in sequential filtrations there should be not too much of a problem with adsorbed inorganic carbon.

Line 312-326: Here, the authors compared the present study with the previous ones. Such comparison should be done in introduction section, because when I read introduction, I felt the novelty of this study is unclear. Response: Hopefully with the changes and additional explanations the manuscript read better now.

Line 330-334: I think hydrogels would also partly pass through the first filter as well as bacterial cells. I cannot understand why the author only mention aggregation regarding hydrogel. Response: Added in the text the possibility of gels passing the first filter and

being retained by the next filter.

Line 335-337: Please describe more detail of the literature; Maske and Garcia (1994). How much amount of carbon was absorbed? Is it possible to explain the results in this study from the amount of carbon absorbed in the literature? Was the seawater used in the literature comparable with the present study? Response: The purpose of the study was to compare filters for primary production measurements using 14C marked phytoplankton. We had no possibility to measure organic carbon.

Line 339-341: The connection between this and past sentence is unclear. What is the reason why the author provided the assumption? Response: Maske and Garcia (1994) found adsorption of organics to inorganic filters immersed in 14C enriched dissolved organic matter, they did not pass the sample through the filters therefore there was no collection of particles on the filter, and no turbulence that could have provoked the formation of gels. For that reason we assume that in general some adsorption of DOC can occur and increase the measured POC, but we consider the quantity of adsorbed DOC to be small compared to the organic aggregates formed or the bacterial biomass. Our assumption is based on the relative increases observed in deep water samples and the capillary experiments.

Line 343-350: The possibility that the authors proposed is not limited to bacteria. I feel strange why they put this information just before the discussion about bacteria. Response: This part was moved to Introduction.

Line 359: What does "this assumption" imply here? Please clarify. Response, changed to: In Figure 8 we used 50% retention of bacteria at every GF/F passage, this estimate follows roughly the bacterial abundance data in Figure 8.

Line 363-371: I also think bacterial cell trapped on 2nd filtration would be small, and carbon content per one cell could be low. Therefore, estimate of carbon in bacterial cells in Fig. 9 would be not reliable, and the discussion based on Fig. 9 is meaningless. Response: As mentioned in we expect our estimate of bacteria biomass in the filtrate to

be over-estimated in Figure 9. We expect the Figure 9 and related discussion to show that the POM found in refiltered samples was only to a small part bacterial biomass.

Line 372-378: Again, less contribution of bacteria is not evidence of aggregation. The authors should provide evidence. Response: By itself the difference is not evidence, but in the context of all the data aggregation is a likely explanation. Bacteria certainly cannot explain that POM2 of deep-water samples is higher than POM1.

Line 385-386: If shear stress during filtration really affect the concentration of TEP, as the author described, interpretation of TEP is difficult. However, the authors did not show clear evidence in this study. Response: TEP data were eliminated from the new manuscript

Line 400-401: The difference in staining between Gum Xanthan and natural gels is not problem, because Gum Xanthan is just a standard compound. Problem is the ratio of TEP-carbon to Gum Xanthan equivalent is variable among samples. Response: TEP data were eliminated from the new manuscript

Line 405-416: I cannot follow the authors' thought in that they described rapid and nonbiogenic phase transition is supported. What is the direct evidence for this finding? In the section from 405-415, they introduce the previous reports, but suddenly said "Our data support âĹij". Here, what is the "our data"? POC, TEP or bacteria? Please explain step by step. Response: I hope our new capillary experiment data convinces the reviewer otherwise.

Line 420-424: The proposed idea of the authors is not sufficiently proved. Response: I hope the new version of the manuscript including our capillary experiment convinces the reviewer otherwise.

Line 441-450: Many uncontrolled factors are considered in this section. If the condition on board has serious effect on the data, this study has serious problem in the viewpoint of reproducibility, because no one can treat sample with same manner in this

study. Response: The text was shortened and changed to: There is no experimental evidence that the change in temperature and pressure that deep water samples would experience could control the aggregation of organics.in a filtrate. Biogenic gel formation can be ruled out because the concentration of microorganisms in the filtrates was much reduced in the filtrates and the time between successive filtration was too short.

Line 444: I think reviewers' comment should not be directly mentioned in the main text. Line 451-458: The authors should carefully interpret the results. Aggregation should be supported by clear evidence. Throughout this manuscript, I cannot find any clear evidence for aggregation. Response: Reviewers comment eliminated, and see text change above. I hope our new capillary experiment data convinces the reviewer otherwise.

Line 459-479: As the author mentioned, effect of turbulence on biogeochemical cycle has been recently attracted phenomenon. However, in this study, they did not provide comment any information on turbulence or shear stress. In addition, I think aggregation in quite short period is suspicious. Response: I hope our new capillary experiment data convinces the reviewer otherwise.
* * *
[Figure]

Figure 1. Sequential filtration of seawater samples; sample identifier is indicated.

[Figure]

[Figure]

Figure 2. 24 replica pairs of POC and PON were tested for reproducibility of the method. Data pairs include in situ samples, cultures and refiltered samples. Multiple replicates were graphed with random x or y assignment. The 95% confidence interval of the type 2 regression (dashed lines) and the confidence interval of the population (dotted line) is indicated (SigmaPlot).

[Figure]

Figure 3. The effect of vapor acidification on POC and PON using precombusted GFF filters wetted with distilled water. Five experiments using five samples each for acid treatment and blanks. Fresh concentrated HCl was used except where noted. 1. The desiccator was cleaned with solvent; 2. Desiccator was cleaned with water and detergent; 3. Desiccator cleaned with solvent. The desiccator top was sealed with grease in 5 and 6. In experiment 3 one acid treated sample is marked as a square because we applied half of the concentration of two duplicates measured together. The dashed line indicates the lower limit of detection.

[Figure]

[Figure]

[Figure]

[Figure]

[Figure]

[Figure]

Figure 6. Log/log scale (POM2/POM1) ratio versus (depth). Both type 1 regressions are significant ($p<0.05$).

[Figure]

Figure 7. Bacterial abundance after filtration step $i$ ($B_i$) normalized to prefiltration abundance ($B_i/B_0$) is indicated for filtrations with double GFF filters and single GFF filtrations. In this graph the abscissa numbers indicate the number of passage ($F_i$) through pre-combusted GFF filters before the bacterial abundance was sampled; i.e. position 0 shows the normalized bacterial abundance in the original sample. $i = 2$ indicates the relative abundance after passage

[Figure]

Figure 8. Measured POC versus the POC calculated to represent the bacterial biomass retained by the GFF filter. Bacterial POC was calculated from bacterial abundance of the sample and a retention efficiency of 50% per GFF passage.

[Figure]

---

## Author Comment (AC2) · 30 Oct 2020

Anonymous Referee #2 The paper by Villaverde et al. presents the results from testing the method for measuring particulate organic matter (POM) in seawater samples, focusing on artifacts associated with filtration. They discuss the implications of these measurements, emphasizing their relevance for conversion of dissolved organic matter to particulate by coagulation processes. The paper presents interesting new ideas that are worth publication, but

the presentation needs to be improved. There is a lot of repetition of some of the ideas that can make reading this numbing.

GENERAL RESPONSE: The reviewers were understandably skeptical of the process of aggregation of dissolved organics promoted by hydraulic stress on the time scale of seconds. To answer doubts we recently did some additional experiments where we compare pre-filtered coastal surface water that was directly re-filtered as in the data reported in the original manuscript (non-stressed) or passed through a capillary (0.5mm ID) and then re-filtered (stressed). In the new Fig. 4 we show the difference between stressed and non-stressed POC and PON. We show that the difference between stressed minus non-stressed POC and PON are significantly higher than zero. It is difficult to compare quantitatively the hydraulic stress exposure of passing through a capillary or through a filter, but the time scales of exposure are similar. The GF/F filters of stressed and non-stressed samples will contain bacterial biomass, adsorbed organics and gels formed during the prefiltration, but the difference between both samples should be due to aggregated dissolved organics formed by passage through the capillary. See details of our simple and easily reproducible experiment in methods and results. As suggested we calculated a lower limit of detection for the POC and PON method and eliminate the data below this limit from figures and interpretation. One exception are the data were we compared the effect of sample acidification. We had included TEP data in the original manuscript submitted because we wanted to make a link to the abundant TEP literature. We eliminated the TEP data from the new manuscript because they did not present a pattern with statistical significance. We left some of TEP related discussion. Hopefully without the TEP data the new manuscript is more concise. The old Fig. 4 (POC and PON with refiltrations) was eliminated, because some of that information is reported in Fig. 5 and to make the manuscript more compact. As suggested by reviewers we changed the Fig. 5 to distinguish the patterns better even when printed in black and white. We added a figure 9 with a conceptual sketch.

SPECIFIC RESPONSE: Issues: Use of the expression "membrane enclosed particles (MEPs)" is a rather peculiar way to refer to non-TEP, non-gel particles, given that it is mixture of fecal pellets, diatom frustules, dead algae, dead animals, dust, ..., as well as bacteria and algal cells. Logan (Logan 1993 L&O 38: 372; Logan et al. 1994. L&O 39: 390) has made similar observations on the collection by glass fiber filters of organisms that are smaller than the ostensible pore sizes. He used classical filtration theory in his analysis. Similar processes should be occurring with the filtration of colloidal organic matter. That is, the actual mechanism for collection may not be the production of larger particles passing through the filters but be related to direct filtration processes on the colloid removal. It would be nice to see what fraction of the "dissolved" gels is removed by each pass through the filters. This would involve providing filtered volumes and DOM concentrations.

Response: Yes, it will be interesting to complete the budget by measuring TOC, POC and DOC, although it does not resolve the question of what is retained on the GF/F filters in the first and subsequent filtrations. Size selective filtration is difficult because of the aggregation of organics with each passing and the associated hydraulic stress. It is possible that a polycarbonate filter produces less hydraulic stress than a depth filter – we have not tried that. Our results leave the question of gel size formed unanswered but documents the principal of the process. In our manuscript we discuss the passage of bacteria through GF/F filters to show that in refiltered samples we have to expect some bacterial biomass. While this does explain partially the organics retained by a second filter, it does not explain that in deep samples the second filter can retain more organics than in the first filter.

Need to have clear separation between Methods, Results, and Interpretation (also known as Discussion) sections. For example, - - L172-174 and L185-190 belong in the Methods section. - - L231-236, L238-239, L248-258 all include comparisons of results with previous literature and belong in the Discussion section. The Interpretation/Discussion section needs to be condensed. Given the relatively few experimental

findings and small amount of data interpretation, having 7 pages for the discussion of a 15 page manuscript is a lot. Response: Since the old Fig. 4 was removed most of this former section 3.2. was eliminated. We did not succeed to reduce the discussion, but hopefully it is more concise.

Stylistic suggestions: 1. Use clear demarcation of new paragraphs. For example, indent the beginning of each paragraph or have a blank line between paragraphs. 2. Use different symbol shapes (+, x, o...) for different data sets rather than the same symbols and different colors. The colors are hard to differentiate, particularly for copies made on black and white printers. 3. Break the really long sentences into more than one. Reading technical papers is difficult enough without having to tease apart complicated sentence structures to understand the arguments. 4. Make the notation consistent throughout the manuscript. For example, L13 has POM2/POMi. Should this not be POM2/POM1? In L115-6, POM1, POC1 and PON1 should be italicized and the 1s should be subscripted. Place a space between the number and the unit when giving data (e.g. L13, 14, 26...). Response: Good suggestions to change the figures; please see the new figures. 'POM2/POMi' was an error now corrected. We tried to shorten the manuscript, but ended up with the same number of pages. We attended to the editorial notes of the reviewer in the attachment

[Figure]

Figure 1. Sequential filtration of seawater samples; sample identifier is indicated.

[Figure]

[Figure]

Figure 2. 24 replica pairs of POC and PON were tested for reproducibility of the method. Data pairs include in situ samples, cultures and refiltered samples. Multiple replicates were graphed with random x or y assignment. The 95% confidence interval of the type 2 regression (dashed lines) and the confidence interval of the population (dotted line) is indicated (SigmaPlot).

[Figure]

Figure 3. The effect of vapor acidification on POC and PON using precombusted GFF filters wetted with distilled water. Five experiments using five samples each for acid treatment and blanks. Fresh concentrated HCl was used except where noted. 1. The desiccator was cleaned with solvent; 2. Desiccator was cleaned with water and detergent; 3. Desiccator cleaned with solvent. The desiccator top was sealed with grease in 5 and 6. In experiment 3 one acid treated sample is marked as a square because we applied half of the concentration of two duplicates measured together. The dashed line indicates the lower limit of detection.

[Figure]

[Figure]

[Figure]

[Figure]

Figure 6. Log/log scale (POM2/POM1) ratio versus (depth). Both type 1 regressions are significant (p<0.05).

[Figure]

Figure 7. Bacterial abundance after filtration step $i$ ($B_i$) normalized to prefiltration abundance ($B_i/B_0$) is indicated for filtrations with double GFF filters and single GFF filtrations. In this graph the abscissa numbers indicate the number of passage ($F_i$) through pre-combusted GFF filters before the bacterial abundance was sampled; i.e. position 0 shows the normalized bacterial abundance in the original sample, $i = 2$ indicates the relative abundance after passage

[Figure]

Figure 8. Measured POC versus the POC calculated to represent the bacterial biomass retained by the GFF filter. Bacterial POC was calculated from bacterial abundance of the sample and a retention efficiency of 50% per GFF passage.

[Figure]

---

## Author Comment (AC3) · 30 Oct 2020

Anonymous Referee #3 This study presented a phenomenon of POM formation in filtered DOM filtrates. Dynamic exchanges between DOM and POM are well-known and can be caused by an array of physical, chemical, and biological factors, including sunlight, pH, cations, phytoplankton, hydrodynamics, etc. It's also related to concentration, composition, and size of DOM and POM. It's an interesting but complicated topic. However, it's unclear in this study what're the detailed organic matter composition and operation conditions of filtration. According to the description in "Methods" section, it seemed that the filtration procedures have been partly done onboard. Photo-flocculation can occur but the sunlight doses onboard and in the lab are supposed to different. Moreover, why did the authors use distilled water instead of ultrapure MQ-H2O during the experiments? Distilled water itself may contain some amount of organic matter. In addition, the authors need to take into account of adsorption and subsequent desorption of retained colloidal DOM. Also, using a second filter to estimate filtration blank is problematic. Below are some specific comments: Throughout the manuscript: 1)add a space between number and unit; 2) change GFF filters to GF/F; 3) subscript numbering of filtration times; 4) pay attention to the punctuation and grammars.

GENERAL RESPONSE: The reviewers were understandably skeptical of the process of aggregation of dissolved organics promoted by hydraulic stress on the time scale of seconds. To answer doubts we recently did some additional experiments where we compare pre-filtered coastal surface water that was directly re-filtered as in the data reported in the original manuscript (non-stressed) or passed through a capillary (0.5mm ID) and then re-filtered (stressed). In the new Fig. 4 we show the difference between stressed and non-stressed POC and PON. We show that the difference between stressed minus non-stressed POC and PON are significantly higher than zero. It is difficult to compare quantitatively the hydraulic stress exposure of passing through a capillary or through a filter, but the time scales of exposure are similar. The GF/F filters of stressed and non-stressed samples will contain bacterial biomass, adsorbed organics and gels formed during the prefiltration, but the difference between both samples should be due to aggregated dissolved organics formed by passage through the capillary. See details of our simple and easily reproducible experiment in methods and results. As suggested we calculated a lower limit of detection for the POC and PON method and eliminate the data below this limit from figures and interpretation. One exception are the data were we compared the effect of sample acidification. We had included TEP data in the original manuscript submitted because we wanted to make a link to the abundant TEP literature. We eliminated the TEP data from the new manuscript because they did not present a pattern with statistical significance. We left

some of TEP related discussion. Hopefully without the TEP data the new manuscript is more concise. The old Fig. 4 (POC and PON with refiltrations) was eliminated, because some of that information is reported in Fig. 5 and to make the manuscript more compact. As suggested by reviewers we changed the Fig. 5 to distinguish the patterns better even when printed in black and white. We added a figure 9 with a conceptual sketch.

SPECIFIC RESPONSE: Line 19 change "Organic particulate matter (POM)" to "Particulate organic matter" Response: Changed

Lines 86-88 This sentence is obscure and need to be re-phrased. Response: The sentence was changed according to some specific suggestions by another reviewer: 'Hydrogels have different physical and chemical properties than MEPs for example, gels are less rigid because they are not contained by membranes or solid material, they are porous, containing large volume portions of water and thus have a specific weight similar to water making them less prone to sink, and sometimes they can float (Mari et al., 2017).'

Lines 101-102 This is speculative. Or please provide reference. Response: In the sentence the word 'show' was changed to 'suggest' : 'Below our data suggest the formation of significant amounts of organic aggregates or micro-gels in filtrates on timescales that are too short for biological production thus suggesting abiotic precursor aggregation.' Hopefully the data from the capillary experiments convince the reviewer

Lines 109-112 Algal cultures are prone to form aggregates within a period of time. Please take this into account. Response: We only used these cultures as producers of EPS, we did not try to relate the physiological condition of the culture to EPS production.

Lines 114 and 122 Correct the degree symbol Response: Corrected

Line 137 "reproducibility o deep...", do you mean "reproducibility of deep..." Response:

Corrected

Line 157 0.45 mm or 0.45 um? Response: Yes, this was an error. It was eliminated together with the other parts related to TEPs

Lines 157-158 What are the grades and brands of these chemicals Response: The TEP part was eliminated

Line 162 Superscript "-1" in the unit Response: Corrected

Line 311 Change "4.1.3" to "4.1.2" and also the following subsection numbers Response: Corrected

Figure 3 "precombusted" or "pre-combusted"? Please be consistent through the manuscript Response: Corrected

Figure 7 missing unit for the filter pore size Response: Figure 7 was eliminated because it showed TEP data

[Figure]

[Figure]

Figure 1. Sequential filtration of seawater samples; sample identifier is indicated.

[Figure]

[Figure]

Figure 2. 24 replica pairs of POC and PON were tested for reproducibility of the method. Data pairs include in situ samples, cultures and refiltered samples. Multiple replicates were graphed with random x or y assignment. The 95% confidence interval of the type 2 regression (dashed lines) and the confidence interval of the population (dotted line) is indicated (SigmaPlot).

[Figure]

Figure 3. The effect of vapor acidification on POC and PON using precombusted GFF filters wetted with distilled water. Five experiments using five samples each for acid treatment and blanks. Fresh concentrated HCl was used except where noted. 1. The desiccator was cleaned with solvent; 2. Desiccator was cleaned with water and detergent; 3. Desiccator cleaned with solvent. The desiccator top was sealed with grease in 5 and 6. In experiment 3 one acid treated sample is marked as a square because we applied half of the concentration of two duplicates measured together. The dashed line indicates the lower limit of detection.

[Figure]

[Figure]

[Figure]

[Figure]

[Figure]

Figure 6. Log/log scale (POM2/POM1) ratio versus (depth). Both type 1 regressions are significant (p<0.05).

[Figure]

Figure 7. Bacterial abundance after filtration step $i$ ($B_i$) normalized to prefiltration abundance ($B_i/B_0$) is indicated for filtrations with double GFF filters and single GFF filtrations. In this graph the abscissa numbers indicate the number of passage ($F_i$) through pre-combusted GFF filters before the bacterial abundance was sampled; i.e. position 0 shows the normalized bacterial abundance in the original sample. $i = 2$ indicates the relative abundance after passage

[Figure]

Figure 8. Measured POC versus the POC calculated to represent the bacterial biomass retained by the GFF filter. Bacterial POC was calculated from bacterial abundance of the sample and a retention efficiency of 50% per GFF passage.

[Figure]